# Framework and Benchmarks for Combinatorial and Mixed-variable Bayesian Optimization

**Kamil Dreczkowski** *†
Imperial College of London
`krd115@ic.ac.uk`

**Antoine Grosnit***
Huawei Noah's Ark Lab
Technische Universität Darmstadt

**Haitham Bou-Ammar**
Huawei Noah's Ark Lab
University College London

## Abstract

This paper introduces a modular framework for Mixed-variable and Combinatorial Bayesian Optimization (MCBO) to address the lack of systematic benchmarking and standardized evaluation in the field. Current MCBO papers often introduce non-diverse or non-standard benchmarks to evaluate their methods, impeding the proper assessment of different MCBO primitives and their combinations. Additionally, papers introducing a solution for a single MCBO primitive often omit benchmarking against baselines that utilize the same methods for the remaining primitives [1–4]. This omission is primarily due to the significant implementation overhead involved, resulting in a lack of controlled assessments and an inability to showcase the merits of a contribution effectively. To overcome these challenges, our proposed framework enables an effortless combination of Bayesian Optimization components, and provides a diverse set of synthetic and real-world benchmarking tasks. Leveraging this flexibility, we implement 47 novel MCBO algorithms and benchmark them against seven existing MCBO solvers and five standard black-box optimization algorithms on ten tasks, conducting over 4000 experiments. Our findings reveal a superior combination of MCBO primitives outperforming existing approaches and illustrate the significance of model fit and the use of a trust region. We make our MCBO library available under the MIT license at `https://github.com/huawei-noah/HEBO/tree/master/MCBO`.

## 1 Introduction

The goal of mixed-variable and combinatorial optimization is to seek optimizers of functions defined over search spaces whose sizes grow exponentially with their dimensions. Applications of this field are ubiquitous, spanning a wide range of domains, such as supply chain optimization [8, 9], vehicle routing [10, 11], machine scheduling in manufacturing systems [12, 13], asset optimization [14, 15], among many others. A general recipe for tackling such tasks involves using the knowledge of domain experts to formalize combinatorial problems within the scope of well-established suited mathematical frameworks, and to apply available heuristic-based solvers. For instance, making the problem compatible with linear, quadratic, nonlinear or integer pro-

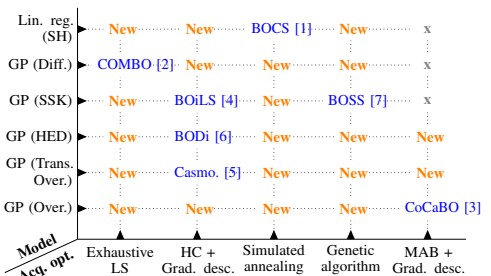

Figure 1: In the MCBO framework, we can build an existing or a new algorithm by choosing a surrogate model and an acquisition function optimizer using a single line of code.

37th Conference on Neural Information Processing Systems (NeurIPS 2023) Track on Datasets and Benchmarks.

---

*These authors contributed equally to this work.

†Work done during an internship at Huawei Noah's Ark Lab in London RC.

gramming solvers, or reducing the combinatorial task at hand to a standard problem such as knapsack [16], traveling salesman [17], or network flow problem [18], allows the application of optimized and scalable off-the-shelf software developed by many companies and academic laboratories [19–22].

Despite numerous successes on many combinatorial tasks, standard heuristics like those mentioned above fall short in vital domains that require the optimization of **black-box objectives**. In those instances, we have, at best, partial a priori knowledge about the characteristics of the optimization objective, making it difficult, if not impossible, to map it to one of the forenamed mathematical frameworks. Although well-established black-box optimization methods such as Simulated Annealing (SA) [23, 24], Genetic Algorithms (GAs) [25], and Evolutionary Algorithms (EAs) [26], as well as online learning methods like Multi-Armed-Bandits (MABs) [27], can still be used to solve such problems, they often fall short at optimizing **expensive-to-evaluate objectives** due to their high sample complexities. Consequently, addressing problems with **expensive-to-evaluate black-box objectives** necessitates the development of **data-driven** and **sample-efficient** solution methodologies.

One promising strategy to handle such objectives is to **1)** efficiently learn a (local) probabilistic model of the black-box function, and **2)** balance exploration and exploitation by leveraging the model's uncertainty. This concept lies at the heart of Mixed-Variable and Combinatorial Bayesian Optimization (MCBO), a machine learning (ML) subfield crucial for achieving efficient optimization with immense potential for solving practical mixed-variable and combinatorial optimization problems.

MCBO algorithms generally have three high-level primitives: a probabilistic surrogate model, an acquisition function, and an acquisition optimizer that can operate in a trust region (TR). As shown in Fig. 1, we can frame many published MCBO algorithms as a specific combination of primitives, illustrating the intrinsic modularity of MCBO. For example, BOiLS [4] uses a Gaussian Process (GP) [28] with the string subsequence kernel (SSK) [29, 30] as its surrogate, and optimizes its acquisition function (expected improvement (EI)) via TR-constrained Hill Climbing [5]. BOSS [7] shares two of its primitives with BOiLS but only differs in using a GA to optimize its acquisition function.

Although many existing MCBO algorithms share some of the primitives they use, it remains unclear which solutions are state-of-the-art for each primitive and which combination of primitives constitutes a state-of-the-art MCBO method. This issue arises due to two factors.

Firstly, MCBO papers often introduce non-diverse and/or non-standard benchmarks to evaluate their proposed methods. The lack of a standardized set of diverse benchmarks makes it difficult to assess the relative performance of the different MCBO primitives and their combinations. As observed in other fields, establishing standardized test domains is crucial for accelerating progress in MCBO. For example, the ImageNet dataset [31], ShapeNet, and MuJoCo [32] facilitated rapid advancements in 2D computer vision, 3D computer vision, and robotics research. Therefore, it is essential to establish a procedural generation of test domains in MCBO, enabling a systematic evaluation of existing methods to inspire the development of novel approaches.

Secondly, papers introducing a solution for a single MCBO primitive often forget to benchmark against baselines that use the same methods for the remaining primitives [1–4], failing to fully highlight the merits of their proposed solution in a controlled setting. This is most likely due to the need for time-consuming and tedious efforts to modify and combine MCBO primitives from existing open-source implementations, as no standardized API exists to allow the interactions among them.

To address the aforementioned problems, we propose a flexible and comprehensive Python framework for MCBO. Our library provides a high-level API for all the MCBO primitives, and implementations of the primitives of several key MCBO baselines. Our framework also features a `BoBuilder` class that enables effortless implementation of existing and novel algorithms by flexibly combining MCBO primitives using a single line of code. In Section 5, we showcase the versatility of our library by implementing seven existing MCBO baselines and 47 novel MCBO algorithms. We evaluate these against five standard black-box optimization baselines on ten tasks, revealing insights into key MCBO design choices that contribute to robust algorithms.

Our library also includes implementations of a wide range of synthetic and real-world mixed-variable and combinatorial benchmarks, covering a broad spectrum of domain dimensionalities and optimization difficulties. These benchmarks include well-known optimization problems, such as the Ackley function [33] and the Pest Control problem [2], and optimization problems that extend

the current application venues of BO, including Antibody Design, RNA inverse folding, and Logic Synthesis optimization.

In the interest of space, throughout this paper we make heavy use of abbreviations. We therefore provide a list of these abbreviations, alongside their definitions, in Table 1 of Appendix A.

## 2 Related Work

**Bayesian Optimisation Frameworks** Existing popular libraries for Bayesian Optimization (BO), such as Spearmint [34], GPyOpt [35], Cornell-MOE [36], RoBO [37], Emukit [38], Dragonfly [39], ProBO [40], and GPFlowOpt [41], offer diverse capabilities and modeling techniques for various optimization tasks. These libraries excel in different areas, including hyperparameter sampling, input warping and parallel optimization [34, 35], multi-fidelity optimization [36–38], and probabilistic programming [40]. However, in contrast to our work, these libraries primarily focus on continuous and discrete search spaces and lack direct support for combinatorial and mixed-variable domains.

Closest to our work is BoTorch [42], a model-agnostic Python library that is integrated with GPyTorch [43], providing efficient and scalable implementations of GPs in PyTorch [44]. Although BoTorch includes the GP kernel for mixed-variable optimization proposed by Wan et al. [5], it primarily focuses on continuous and discrete search spaces and does not provide direct built-in support for combinatorial formulations. In contrast, our framework is designed to tackle the unique challenges of BO in combinatorial and mixed-variable spaces and extends the capabilities of existing libraries by incorporating surrogate models and acquisition optimization techniques tailored to these domains.

Furthermore, our framework offers an unprecedented level of ease and flexibility in implementing BO algorithms by **mixing-and-matching** implemented BO primitives. With just a **single line of code**, users can combine different surrogate models, acquisition functions, and optimization methods. In contrast, other libraries would require time-consuming manual implementation processes to make components match.

**Mixed-variable and Combinatorial Optimization Benchmarks** Existing popular benchmarks for mixed-variable and combinatorial problems include the 24 synthetic mixed-integer test functions [45] that are part of the COCO [46] benchmarking suite. Our framework extends some of these functions, including the sphere, rotated hyper ellipsoid, Rastrigin, and Rosenbrock functions, to accommodate arbitrary dimensional domains and combinations of variable types. Moreover, we introduce additional synthetic functions that uphold the same generalization principles.

Distinguishing our benchmarking suite from benchmarks such as HPOBench [47], HPO-B [48], and YAHPO Gym [49], which primarily focus on a single domain such as hyperparameter optimization, our framework takes a more comprehensive perspective. Our suite encompasses a broader spectrum of tasks, spanning hyperparameter tuning, antibody design, Electronic Design Automation, RNA inverse folding, along with the classical synthetic tasks. This multi-domain nature underpins our conviction that benchmarking on diverse tasks that cover a broad spectrum of problem types and complexities strengthens the robustness and reliability of findings. We hold firm to the notion that this multifaceted approach leads to more generalisable conclusions than scenarios where only a single family of tasks is used for benchmarking, which can still be useful for important black-box families.

## 3 Mixed-Variable and Combinatorial Bayesian Optimization

BO is a sequential model-based technique to efficiently optimize a black-box function $f(\cdot)$ over a search space $\mathcal{X}$. Due to the black-box nature of $f(\cdot)$, we can only evaluate it at input locations $\boldsymbol{x} \in \mathcal{X}$ to get (noisy) outputs $y \in \mathbb{R}$, such that $\mathbb{E}[y|f(\boldsymbol{x})] = f(\boldsymbol{x})$. BO tackles global optimization problems by iteratively repeating two steps. At iteration $i$, it first *suggests* a point $\boldsymbol{x}_i$ to evaluate, and in a second step, the learner *observes* $y_i$, by evaluating the black-box at $\boldsymbol{x}_i$. To enable sample efficiency, the suggestion of $\boldsymbol{x}_i$ typically involves learning a (local) probabilistic surrogate model of $f(\cdot)$ from the set of already observed points, and is followed by the optimization of an acquisition function $\alpha(\cdot)$ trading-off exploration vs exploitation. We highlight both steps in Alg. 1, and depict a generic BO loop operating for a total budget of $T_{\max}$.

### 3.1 The Surrogate Model

Optimizing expensive black-box functions requires modeling the objective to enable efficient search by prioritizing informed decision-making over blind function evaluations [34]. When dealing with mixed-variable and combinatorial formulations, various surrogate models are available, including Bayesian Linear Regression [50], Random Forests [51], Tree-Structured Parzen Estimators [52], and Bayesian Neural Networks [53]. Still, GPs [28] remain the most widely adopted models in the literature due to their tractability, sample efficiency, and capacity to maintain calibrated uncertainties [28, 34, 39, 54, 55], and therefore our library mostly focuses on their support.

---

**Algorithm 1:** Generic BO Algorithm

---

**Input:** Initial dataset
$\mathcal{D}_{n_0} = \{\boldsymbol{x}_j, y_j\}_{j=1}^{n_0}$, budget
$T_{\max}$, black-box $f$.

**for** $i = n_0 + 1, ..., T_{\max}$ **do**

> Learn a surrogate model $\mathcal{M}(\cdot|\mathcal{D}_{i-1})$ of $f$.
> Use $\mathcal{M}(\cdot|\mathcal{D}_{i-1})$ to define acquisition function $\alpha(\boldsymbol{x}|\mathcal{D}_{i-1})$.
> Optimize $\alpha(\boldsymbol{x}|\mathcal{D}_{i-1})$ to get $\boldsymbol{x}_i$.
> Get $y_i$ by evaluating $f$ at $\boldsymbol{x}_i$.
> Set $\mathcal{D}_i \leftarrow \mathcal{D}_{i-1} \cup \{\boldsymbol{x}_i, y_i\}$.

**Output:** $\boldsymbol{x}^\star = \arg\min_{\boldsymbol{x}_i \in \mathcal{D}_{T_{\max}}} y_i$.

---

A GP is a non-parametric model that represents the prior belief about a black-box function as a distribution over functions and updates this distribution as new observations become available, resulting in a posterior distribution. A GP is fully defined by its mean function $m(\cdot)$ and kernel function, $k_{\boldsymbol{\theta}}(\cdot, \cdot)$, where $\boldsymbol{\theta}$ are the kernel hyperparameters [28, 54]. The mean function captures the overall trend and bias of the modeled function, while the kernel function characterizes the correlation between function values at different input locations. Specifically, the kernel function, $k_{\boldsymbol{\theta}}(\boldsymbol{x}, \boldsymbol{x}')$, expresses our assumptions regarding the smoothness and periodicity of the modeled function, as it corresponds to the covariance between pairs of function values $\text{Cov}(f(\boldsymbol{x}), f(\boldsymbol{x}'))$.

When modeling black-box functions defined over combinatorial spaces, one commonly used kernel function is the Overlap kernel [3, 56, 57], defined using Kronecker delta function $\delta(\cdot, \cdot)$ as

$$k_{\boldsymbol{\theta}}^{\text{O}}(\boldsymbol{x}, \boldsymbol{x}') = \frac{\sigma}{d} \sum_{p=1}^{d} \lambda_p \, \delta\left(\boldsymbol{x}[p], \boldsymbol{x}'[p]\right), \tag{1}$$

with $\boldsymbol{\theta} = (\sigma, \lambda_1, \dots, \lambda_d) \in \mathbb{R}_+^{d+1}$, where $\sigma$ represents the kernel variance, $d$ is the dimensionality of $\boldsymbol{x}$, and $\lambda_p$ denotes the Automatic Relevance Determining (ARD) [58] length scale for the $p^{\text{th}}$ variable. In combinatorial problems, $\boldsymbol{x} \in \{c_1^1, ..., c_{N_1}^1\} \times ... \times \{c_1^M, ..., c_{N_M}^M\}$, where $\{c_1^i, ..., c_{N_i}^i\}$ is the set defining all the possible elements that the $i^{\text{th}}$ combinatorial variable can assume, $N_i$ is the cardinality of this set, and $M$ is the total number of combinatorial variables. In this context, the Overlap kernel measures the extent to which variables in the two input vectors share the same categories.

A related kernel is the Transformed Overlap (TO) kernel [5] applying the exponential function to the output of the Overlap kernel. This transformation enhances the kernel's expressive power, enabling it to model more complex functions [5]. Other kernels tailored for combinatorial inputs include the SSK [29, 30], the Diffusion kernel (Diff.) [2], and the Hamming embedding via dictionary kernel (HED) [6], which are all supported in our framework.

Given a dataset $\mathcal{D} = \{\boldsymbol{x}_i, y_i\}_{i=1}^n$ with Gaussian-corrupted observations $y_i = f(\boldsymbol{x}_i) + \epsilon_i$, with $\epsilon_i \sim \mathcal{N}(0, \sigma^2)$, and given a GP prior, the posterior of the black-box function value at a test point $\boldsymbol{x}_{\text{test}}$ denoted as $f(\boldsymbol{x}_{\text{test}})|\mathcal{D}, \boldsymbol{\theta}$, is also a Gaussian distribution $\mathcal{N}\left(\mu_{\boldsymbol{\theta}}(\boldsymbol{x}_{\text{test}}), \sigma_{\boldsymbol{\theta}}^2(\boldsymbol{x}_{\text{test}})\right)$. For brevity, we defer to Appendix B the analytic expressions for $\mu_{\boldsymbol{\theta}}(\boldsymbol{x}_{\text{test}})$ and $\sigma_{\boldsymbol{\theta}}^2(\boldsymbol{x}_{\text{test}})$ along with the method to learn the optimal kernel hyperparameters $\boldsymbol{\theta^*}$ by minimizing the negative log-likelihood.

### 3.2 The Acquisition Function

The acquisition function plays a crucial role in BO as it approximates the utility of evaluating the black-box function at a specific input $\boldsymbol{x} \in \mathcal{X}$. Since the black-box function is unknown, the acquisition function considers both the estimated value of the objective function and its associated uncertainty. This enables the acquisition function to effectively balance exploration of the search space, gathering more information about the underlying objective function and exploiting currently promising regions likely to contain the optimal solution. Ultimately, the acquisition function is a criterion for selecting the next point to evaluate in the optimization process.

The Expected Improvement (EI) [59, 60] acquisition function measures the utility of new query points by evaluating the expected gain compared to the function values observed so far, considering

the uncertainty of the model's posterior. During each iteration $i$ of the optimization process, let $y^\star_{1:i-1}$ denote the best black-box function value in the dataset $\mathcal{D}_{i-1}$. The EI function is defined as:

$$\alpha^{(\text{EI})}(\boldsymbol{x}) = \mathbb{E}_{f(\boldsymbol{x})|\mathcal{D}_{i-1},\boldsymbol{\theta}^\star}\left[\max\{y^\star_{1:i-1} - f(\boldsymbol{x}), 0\}\right],$$

where the expectation is taken with respect to the posterior of a trained model. Our framework supports EI as well as other popular acquisitions such as Probability of Improvement (PI) [61], lower confidence bound (LCB) [62], and Thompson sampling (TS) [63], all described in Appendix B.3.

## 3.3 Acquisition Optimization Methods

The acquisition optimization method plays a vital role in BO as it determines the next evaluation point $\boldsymbol{x}_i$ by optimizing the acquisition function $\alpha(\boldsymbol{x}|\mathcal{D}_{i-1})$. In general, maximizing the acquisition function to find $\boldsymbol{x}_i$ is an optimization problem defined globally over the entire search space:

$$\boldsymbol{x}_i = \arg\max_{\boldsymbol{x} \in \mathcal{X}} \alpha(\boldsymbol{x}|\mathcal{D}_{i-1}). \tag{2}$$

When dealing with a continuous search space, optimizing the acquisition function is relatively straightforward and can be performed by using off-the-shelf first or second-order optimization methods with random restarts (refer to [64] for a comprehensive study on this topic). However, when dealing with combinatorial inputs, the absence of a defined gradient hinders the direct application of gradient-based optimization methods [65]. To address this challenge, various zero-order methods have been proposed for maximizing the acquisition function in combinatorial spaces. These methods encompass a range of global optimization algorithms and heuristics, including Hill Climbing (HC) [5], exhaustive Local Search (LS) [2], SA [1], GA [7], and MAB [3], which we include in our library.

These acquisition optimizers are often applied as global optimization algorithms to solve Equation 2. However, just like in continuous spaces, when the dimensionality of the search space is high, the surrogate may struggle to accurately model the black box over the entire search space. Taking inspiration from related work in BO for continuous spaces [66], some recent combinatorial BO algorithms, including Casmopolitan [5], introduce the notion of a TR to constrain the acquisition optimization procedure in the context of MCBO. In Casmopolitan, the TR around the best input found so far in the current trust region, $\boldsymbol{x}^{\text{TR}}$, is defined as

$$\text{TR}(\boldsymbol{x}^{\text{TR}}) = \{\boldsymbol{x} \in \mathcal{X} \text{ s.t. } d_h(\boldsymbol{x}^{\text{TR}}_h, \boldsymbol{x}_h) \leq L_h \text{ and } d_{L_n}(\boldsymbol{x}^{\text{TR}}_n, \boldsymbol{x}_n) \leq 1\}, \tag{3}$$

where $\boldsymbol{x}^{\text{TR}}_h$ and $\boldsymbol{x}_h$ are vectors containing all the combinatorial inputs in $\boldsymbol{x}^{\text{TR}}$ and $\boldsymbol{x}$ respectively, $d_h(\cdot, \cdot)$ is the Hamming distance [5] and $L_h$ is the size of the TR for combinatorial variables. Similarly, $\boldsymbol{x}^{\text{TR}}_n$ and $\boldsymbol{x}_n$ are vectors containing all the numeric and discrete variables in $\boldsymbol{x}^{\text{TR}}$ and $\boldsymbol{x}$, respectively, $d_{L_n}(\cdot, \cdot)$ is the maximum of the component-wise distance for numeric and discrete variables normalized by dimensional length scales $L_n \in \mathbb{R}^{\dim(\boldsymbol{x}_n)}_+$ for the numeric and discrete variables [5].

## 4 The MCBO Framework

To facilitate code reusability and enable benchmarking on a standardized set of tasks, we introduce the MCBO framework, whose source code is open-source under the MIT license. Our ready-to-use software provides **1)** An API for defining BO primitives, **2)** Multiple primitives from key existing MCBO baselines, **3)** A BO constructor to effortlessly combine primitives into new algorithms, **4)** A variety of mixed-type and combinatorial BO and non-BO baselines, **5)** A suite of standard and novel mixed-type and combinatorial benchmarks, and **6)** An API for defining novel optimization problems.

We structure the MCBO framework in a modular fashion to allow for the rapid prototyping of new solutions by combining existing BO primitives. We build MCBO on top of PyTorch [44] to make it compatible with the rich ecosystem of code developed for training various regression models. The entire library structure naturally revolves around seven Python classes; the `SearchSpace`, `TaskBase`, `ModelBase`, `AcqBase`, `AcqOptimizerBase`, `OptimizerBase`, and the `BoBase` class.

## 4.1 Defining Optimization Problems

To frame the optimization of a black-box within our framework, we need to create a task class (inheriting from `TaskBase`) that implements two methods: `get_search_space_params` and `evaluate`. The latter implements the call to the black-box, while the former provides the list of variables that compose the optimization domain of the black-box. We build a search space (an instance of `SearchSpace`) by providing this list of variables with their names, types, and any additional information needed, such as the categories they can take for categorical variables. As an example, consider a combinatorial optimization problem whose search space contains five categorical variables (with categories `"A"`, `"B"`, `"C"`, and `"D"`) and for which the black-box function is already defined as `black_box_script` in a python script. In the MCBO framework, we can create the corresponding task as follows:

```python
class BlackBox(TaskBase):

    def get_search_space_params(self) -> List[Dict[str, Any]]:
        categories = ['A', 'B', 'C', 'D']
        params = [{'name': f'x{i}', 'type': 'nominal', 'categories': categories}
                  for i in range(5)}]
        return params

    def evaluate(self, x: pd.DataFrame) -> np.ndarray:
        return black_box_script(x)

black_box = BlackBox()
search_space = black_box.get_search_space()  # get an instance of SearchSpace
```

## 4.2 Surrogate Models, Acquisition Functions, and Acquisition Optimizers

The MCBO framework includes three core BO primitives: surrogate models, acquisition functions, and acquisition optimizers. **1)** Our implementation features several pre-implemented surrogate models, such as a GP with the string subsequence kernel (SSK) [4, 7], overlap kernel (O) [3], transformed-overlap kernel (TO) [5], diffusion kernel (Diff.) [2], dictionary-based kernel (HED) [6], mixture kernel [5], and linear regression [50] with the Horseshoe prior [67] using maximum likelihood, maximum a posteriori, and Bayes estimation (LHS) [1]. **2)** On the acquisition function side, we include the widely used expected improvement [59, 60], probability of improvement [61] and lower confidence bound [62] for GPs, and Thompson sampling [63] for LHS. **3)** Furthermore, our library includes various acquisition optimizers, such as exhaustive Local Search [2], Genetic Algorithm [7], Simulated Annealing [1], and interleaved search (IS) alternating between Hill Climbing or Multi-armed Bandit for combinatorial variables and gradient-descent steps for numeric variables as developed for CoCaBO [3] and Casmopolitan [5]. Moreover, we generalize the implementation of these acquisition optimizers so that **all of them support TR-constrained acquisition optimization**, extending [5, 68], and make them **handle cheap-to-compute input domain constraints** via rejection sampling.

## 4.3 Defining MCBO Algorithms

With our framework, defining MCBO algorithms is effortless. By specifying the IDs of the surrogate model, acquisition function, acquisition optimizer, and TR manager in the `BoBuilder` class constructor, the corresponding BO primitives are automatically retrieved. In an optimization loop, the optimizer created with `BoBuilder` handles model fit, acquisition optimization, and TR adjustments. Thanks to the `BoBuilder` class, we can easily **mix-and-match** BO primitives from existing algorithms, allowing the implementation of novel MCBO algorithms with just **one line of code per algorithm**. We demonstrate the versatility of this approach by implementing 47 novel MCBO algorithms and evaluating them on a set of ten tasks in Section 5. As an example, we showcase on the next page how to use `BoBuilder` to construct the Casmopolitan algorithm [5] and apply it to optimize a generic black-box function for a specified budget.

### 4.4 Baselines

**MCBO baselines** We leverage the aforementioned BO primitives to implement seven existing MCBO algorithms: Casmopolitan [5], BOiLS [4], COMBO [2], CoCaBO [3], BOSS [7], BOCS [1], and BODi [6]. Furthermore, we explore the remaining combinations of implemented surrogate models and acquisition optimizers, including trust-region-based optimizers, implementing 47 additional novel MCBO algorithms (See Fig. 1). In Section 5, we benchmark the performance of these 54 MCBO algorithms and a set of five standard non-BO baselines that we describe in the following paragraph.

**Black-Box Optimization Baselines** To facilitate benchmarking against non-BO optimization methods, we include the following baselines in our library: Random Search (RS) [69], Hill Climbing (HC) [70], GA [25], SA [23, 24], and MAB [27]. We make them inherit from the `OptimizerBase` class, ensuring a consistent API with MCBO solvers with the use of `suggest` and `observe` methods.

```python
bo_builder = BoBuilder(
    model_id='gp_to',
    acq_opt_id='is',
    acq_func_id='ei',
    tr_id='basic',
    **kwargs
)  # Corresponds to Casmopolitan

opt = bo_builder.build_bo(
    search_space=search_space,
    n_init=20
)

for _ in range(budget):
    x_next = opt.suggest()
    y_next = black_box(x_next)
    opt.observe(x_next, y_next)

print(opt.best_x, opt.best_y)
```

### 4.5 Available Benchmarks

We include diverse benchmarks, enabling a systematic evaluation of MCBO methods across various optimization domains, dimensionalities, and difficulties. The benchmarks encompass both synthetic and real-world tasks. While we briefly overview the available benchmarks here, we provide a more detailed description, including specific constraints, dimensionalities, supported domains, and implementation details in Appendix C.

The synthetic benchmark suite offers a controlled environment for evaluating the performance of MCBO methods, featuring the 21 Simon Fraser University (SFU) test functions [33] that generalize to $d$-dimensional domains, extended to handle continuous, discrete, and nominal variables, as well as combinations of these variable types. Additionally, we include the pest control task [2] that presents a challenging optimization landscape with high-order interactions.

For real-world tasks, the benchmark suite includes logic synthesis optimization [71], antibody design [72], RNA inverse folding [73], and hyperparameter tuning of ML models. The logic synthesis benchmarks optimize Boolean circuits represented by And-Inverter Graphs (AIG) and Majority-Inverter Graphs (MIG) [74]. The antibody design task focuses on optimizing the CDRH3 region of antibodies for binding to a specific antigen. RNA inverse folding aims to find RNA sequences that fold into a target secondary structure. Hyperparameter optimization tasks involve tuning the parameters of XGBoost [75] on the MNIST dataset [76] and $\nu$-Support Vector Regression ($\nu$-SVR) [77] on the UCI slice dataset [78] for which we do feature selection along with the hyperparameter tuning as in [6].

## 5 Experiments

To illustrate the capacity of our library, we conduct experiments tackling the following questions:

1. Which implemented surrogate model and acquisition optimizer performs best?
2. Does incorporating a TR constraint improve the performance of MCBO?
3. Which implemented Combinatorial and Mixed-variable algorithm is the most robust?

We exploit the high modularity and flexibility of our framework to easily instantiate a total of 48 combinatorial BO algorithms using the `BoBuilder` class, and compare their performance on six combinatorial tasks, including Ackley-20D, pest control, sequence of operators tuning for AIG and MIG logic synthesis, antibody design, and RNA inverse fold task. We also run five non-BO baselines - RS, HC, GA, SA, and MAB - on these six combinatorial tasks. Similarly, we use `BoBuilder` to implement 24 mixed-variable BO methods and evaluate them on four mixed-variable

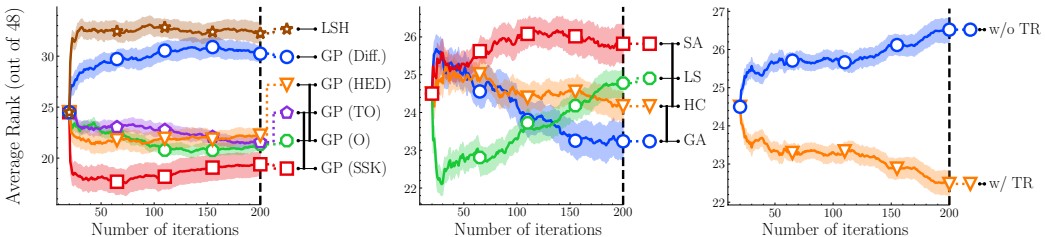

Figure 2: Evolution of the average rank of combinatorial BO algorithms across six tasks, aggregated by surrogate model (left), acquisition optimizer (center), and use of a trust-region (right).

tasks, including Ackley-53D, XGBoost hyperparameter tuning, $\nu$-SVR tuning with feature selection, and logic synthesis flow optimization for AIGs. We benchmark the mixed-variable BO algorithms against four non-BO baselines, RS, HC, GA, and SA. Each experiment is repeated across fifteen random seeds, resulting in over $4,000$ individual experiments.

## 5.1 Experimental Procedure

To ensure a fair and consistent comparison, we follow a standardized experimental design. For a given task and random seed, each BO algorithm suggests the same set of 20 uniformly sampled points and observes the corresponding black-box function values. For the next 180 steps, each algorithm suggests a new point to evaluate based on its surrogate model and acquisition function optimization process. We therefore query the black-box function 200 times per algorithm and per seed. For a given black-box evaluation budget $i = 1, \ldots, 200$, we denote by $y_i^*$ the best value attained so far, $y_i^* = \min_{1 \le j \le i} y_j$[3]. As the scales of the black-box values can differ drastically from one task to another, we compare the optimizers' performance across tasks by considering their ranks based on the $y_i^*$s. To analyze the impact of some primitives, We can then aggregate the ranks across tasks, random seeds, and the BO primitives not under investigation. For example, when assessing surrogate models, we average the rank of the optimizers sharing the same surrogate across tasks, random seeds, acquisition functions, and acquisition optimizers. In figures displaying the evolution of ranks, we show the mean rank in solid line, and the standard error with respect to tasks and seeds as a shaded area. We conduct a Friedman test to see if we can reject the hypothesis that all the methods' performances are equal, and if it is not the case, we add black vertical lines to connect the algorithms whose mean rank differences are smaller than the length of the critical interval given by the post-hoc Wilcoxon signed-rank test.

## 5.2 Hardware

We run our experiments on two machines with 4 Tesla V100-SXM2-16GB GPUs and an Intel(R) Xeon(R) CPU E5-2699 v4 @ 2.20GHz with 88 threads, which allowed us to parallelize over tasks and seeds. It took us approximately two weeks to run the set of experiments described in this paper.

## 5.3 Analyzing Combinatorial BO Design Choices

**Surrogate model** We observe in Fig. 2 (left) that BO methods with LSH or GP (Diff.) surrogate significantly underperform compared to those based on GP (HED), GP (O), GP (TO), and GP (SSK). However, from the well-performing surrogate models, no single model outperforms the others significantly, indicating that certain models are better suited for specific types of black-box functions as we will investigate in Section 5.4.

**Acquisition function optimizers** Fig. 2 (center) shows that exhaustive LS performs best at very low budgets. But as the number of step increases, GA delivers superior mean performance. This suggests that the high exploitation offered by LS is beneficial when few suggestions are allowed, but the less myopic approach of GA achieves a better exploration-exploitation trade-off as budget increases.

---

[3]Reporting regret is not possible for the real-world tasks whose minima are unknown.

**Trust region (TR)** Finally, Fig. 2 (right) shows that, on average, methods working with a dynamic TR to fit local models and constrain acquisition optimization provide consistently better suggestions compared to global approaches. This confirms and extends the findings of Wan et al. [5].

## 5.4 Which is the Most Robust Combinatorial Algorithm?

In this section, we consider each mix-and-match combinatorial BO algorithm individually (without aggregating ranks by primitives), and compare them to non-BO baselines. We first order all the algorithms based on their average ranks across ten seeds and 6 tasks, and we show on Fig. 3 the performance of the 6 known BO solvers, the 5 non-BO baselines, and the 2 (out of 42) new mix-and-match methods achieving the lowest rank.

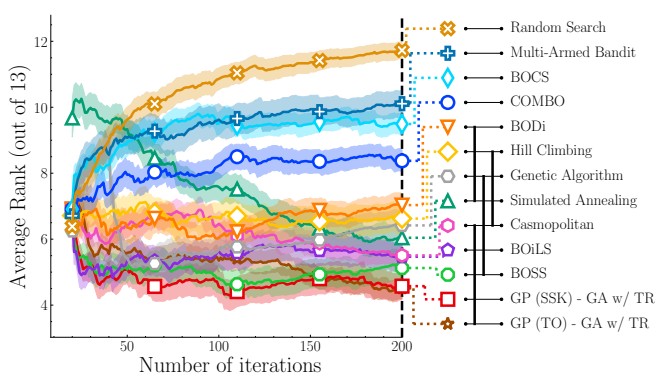

Figure 3: Average rank of known combinatorial algorithms and the two best mix-and-match BO.

We observe that only the two new MCBO algorithms achieve a statistically significant average rank improvement compared to the GA, and SA black-box optimization baselines. These two algorithms utilize a GP (SSK) or GP (TO) surrogate model with a GA and trust region constraint for acquisition optimization. Notably, these algorithms are novel and their design choices align with our conclusions from Section 5.3. However, it is important to highlight that the difference in rank between these algorithms and Casmopolitan [5], BOSS [7], and BOiLS [4] is not statistically significant. This can be attributed to the significant role of model fit (see Section 5.4) in BO and the averaging of results across diverse tasks, where different surrogate models may be optimal.

For each BO optimizer and the three best performing non-BO baselines, we show on Fig. 4 the evolution of the best objective values attained. We note that on two tasks (Antibody design and RNA inverse fold), a non-BO algorithm (SA and GA respectively) outperforms all BO solvers, highlighting the need for further development of combinatorial BO techniques when it comes to solving real-world tasks. Nevertheless, BO solvers are still the best on a majority of tasks, though we observe that no single optimizer achieves the lowest objective value across all tasks. We investigate the variability of BO performance in the next paragraph.

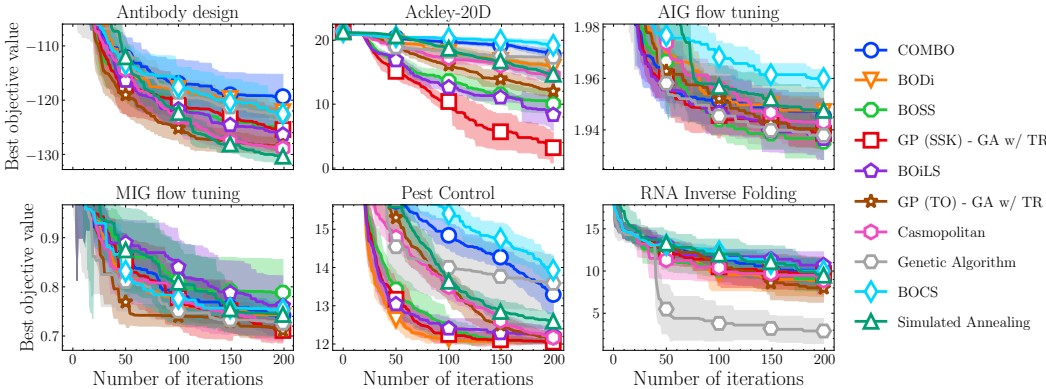

Figure 4: Evolution of best observed value on each combinatorial task achieved by the known combinatorial optimizers and the two best mix-and-match BO solvers built with MCBO framework. In all tasks, the aim is to minimize the black-box. Hence, the lower the best observed value the better.

**Model fit and BO performance**    As underlined in Section 3.1, the choice of kernel impacts the GP modeling capacity, which could explain the variability of BO performance on the different black-box functions. To assess the relation between kernel choice and BO performance, we measure the Pearson correlation between the quality of a GP model fit, and the quality of the objective value attained after 200 iterations of BO equipped with the same surrogate. We get the quality of a GP fit on a given task and seed by conditioning the GP on the first 150 points $\{(\boldsymbol{x}_i, y_i)\}_{i=1}^{150}$ coming from our GA run and computing the log-likelihood of the last 50 black-box values $\{y_i\}_{i=150}^{200}$ under the GP prediction at $\{\boldsymbol{x}_i\}_{i=150}^{200}$. Fixing the acquisition optimizer and the use of TR, we collect for all tasks and seeds the GP log-likelihood and the BO performance when using SSK, TO, O, HED and diffusion kernel. Given a task and a seed, the set of points coming from the GA run are the same to limit a source of noise in the performance comparison. As expected, Fig. 5 shows a positive correlation between BO performance and the capacity of its surrogate model to fit the black-box. The correlation is weaker for local acquisition optimizers (LS) than more explorative ones (GA).

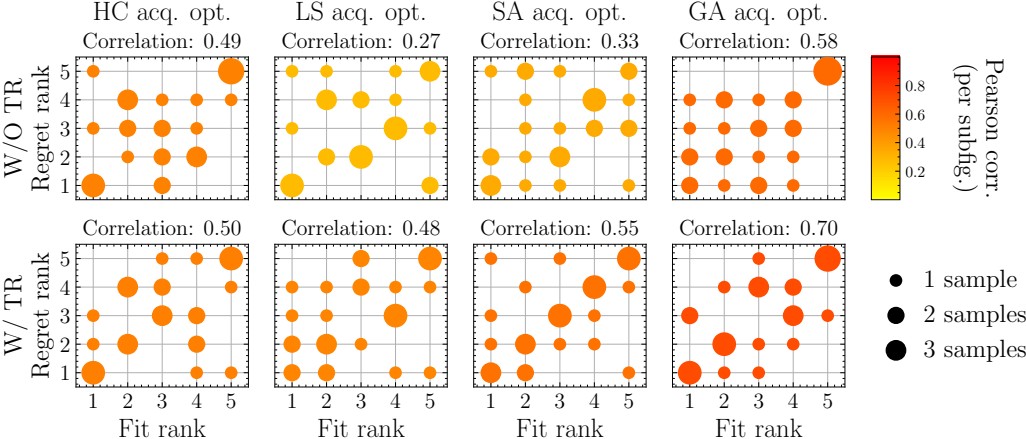

Figure 5: Pearson correlation between the quality of a surrogate fit, and BO regret, split by acquisition function and use of a TR.

## 5.5   Mixed-variable optimization

Due to space limitations, we defer the analysis of results obtained on mixed tasks to Appendix D. This analysis reveals that a new mixed-variable BO algorithm, which combines a GP (Matérn-5/2 and HED kernel) surrogate model with a GA acquisition optimizer and a trust region constraint, achieves a statistically significant better rank averaged across all the considered tasks.

## 6   Conclusion

This work introduces a modular and flexible framework for MCBO, addressing the need for systematic benchmarking and standardized evaluation in the field. Leveraging this framework, we implement a total of 48 combinatorial and 24 mixed-variable BO algorithms with just a single line of code per algorithm. Benchmarking these algorithms on a set of 10 tasks, which is a subset of the tasks available in our framework, we conduct over 4000 individual experiments. Through these experiments, we provide insights into the choice of surrogate model based on its fit, the selection of acquisition optimizer, and the use of a trust region constraint to constrain acquisition optimization.

It is important to note that these experiments represent only a fraction of the possibilities our framework offers. We aim to demonstrate the ease and potential of our MCBO framework, encouraging further analyzes and the development of new primitives. We also invite MCBO developers to integrate their novel options into our flexible codebase, enabling easy mixing and matching of algorithms and facilitating benchmarking across various synthetic and real-world tasks with minimal effort.

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

# Appendix A    Lexicon

In Table A we include a full list of the abbreviations used in this paper, alongside their definitions.

Table 1: A full list of the abbreviations used in this paper alongside their definitions

| Abbreviation | Definition |
|---|---|
| AIG | And-Inverter Graphs |
| API | Application Programming Interface |
| ARD | Automatic Relevance Determining |
| BO | Bayesian Optimization |
| Diff. | Diffusion kernel |
| EA | Evolutionary Algorithm |
| EI | Expected Improvement |
| GA | Genetic Algorithm |
| GP | Gaussian Process |
| HC | Hill Climbing |
| HED | Hamming embedding via dictionary kernel |
| IS | Interleaved Search |
| LCB | Lower Confidence Bound |
| LHS | Linear Regression with the Horseshoe Prior |
| LS | Exhaustive Local Search |
| MAB | Multi-Armed-Bandit |
| MCBO | Mixed-variable and Combinatorial Bayesian Optimisation |
| MIG | Majority-Inverter Graphs |
| ML | Machine Learning |
| O | Overlap Kernel |
| PI | Probability of Improvement |
| RNA | Ribonucleic Acid |
| RS | Random Search |
| SA | Simulated Annealing |
| SFU | Simon Fraser University |
| SSK | String Subsequence Kernel |
| TO | Transformed Overlap |
| TR | Trust Region |
| TS | Thompson sampling |
| $\nu$-SVR | $\nu$-Support Vector Regression |

# Appendix B    Bayesian Optimization

## B.1    Inference with Gaussian Processes

Given a dataset $\mathcal{D} = \{\boldsymbol{x}_i, y_i\}_{i=1}^n$, and assuming Gaussian-corrupted observations $y_i = f(\boldsymbol{x}_i) + \epsilon_i$, where $\epsilon_i \sim \mathcal{N}(0, \sigma^2)$, the joint probability distribution over the observed data and an arbitrary test input $\boldsymbol{x}_{\text{test}}$ can be written as:

$$\begin{bmatrix} \mathbf{y}_{1:n} \\ f(\boldsymbol{x}_{\text{test}}) \end{bmatrix} \Big| \boldsymbol{\theta} \sim \mathcal{N} \left( \mathbf{0}, \begin{bmatrix} \mathbf{K}_{\boldsymbol{\theta}} + \sigma^2 \mathbf{I} & \mathbf{k}_{\boldsymbol{\theta}}(\boldsymbol{x}_{\text{test}}) \\ \mathbf{k}_{\boldsymbol{\theta}}^{\mathsf{T}}(\boldsymbol{x}_{\text{test}}) & k_{\boldsymbol{\theta}}(\boldsymbol{x}_{\text{test}}, \boldsymbol{x}_{\text{test}}) \end{bmatrix} \right). \tag{4}$$

Here, we assume a zero-mean GP prior and use $\mathbf{y}_{1:n}$ to represent the vector of all outputs, i.e., $\mathbf{y}_{1:n} = [y_1, \ldots, y_n]^{\mathsf{T}}$. The block-covariance matrix in the equation above is defined as:

$$\mathbf{K}_{\boldsymbol{\theta}} = \mathbf{K}_{\boldsymbol{\theta}}(\boldsymbol{x}_{1:n}, \boldsymbol{x}_{1:n}) \in \mathbb{R}^{n \times n}, \quad \text{such that } [\mathbf{K}_{\boldsymbol{\theta}}(\boldsymbol{x}_{1:n}, \boldsymbol{x}_{1:n})]_{k,\ell} = k_{\boldsymbol{\theta}}(\boldsymbol{x}_k, \boldsymbol{x}_\ell) \quad \forall (k, \ell)$$

$$\mathbf{k}_{\boldsymbol{\theta}}(\boldsymbol{x}_{\text{test}}) = \mathbf{k}_{\boldsymbol{\theta}}(\boldsymbol{x}_{1:n}, \boldsymbol{x}_{\text{test}}) \in \mathbb{R}^{n \times 1}, \quad \text{such that } [\mathbf{k}_{\boldsymbol{\theta}}(\boldsymbol{x}_{1:n}, \boldsymbol{x}_{\text{test}})]_k = k_{\boldsymbol{\theta}}(\boldsymbol{x}_k, \boldsymbol{x}_{\text{test}}) \quad \forall k.$$

By conditioning the joint distribution, we can derive the posterior distribution for predicting $f(\boldsymbol{x}_{\text{test}})$ [28], resulting in $f(\boldsymbol{x}_{\text{test}}) | \boldsymbol{x}_{\text{test}}, \mathcal{D}, \boldsymbol{\theta} \sim \mathcal{N}\left(\mu_{\boldsymbol{\theta}}(\boldsymbol{x}_{\text{test}}), \sigma_{\boldsymbol{\theta}}^2(\boldsymbol{x}_{\text{test}})\right)$, where:

$$\mu_{\boldsymbol{\theta}}(\boldsymbol{x}_{\text{test}}) = \mathbf{k}_{\boldsymbol{\theta}}^{\mathsf{T}}(\boldsymbol{x}_{\text{test}})(\mathbf{K}_{\boldsymbol{\theta}} + \sigma^2 \mathbf{I})^{-1} \mathbf{y}_{1:n}$$

$$\sigma_{\boldsymbol{\theta}}^2(\boldsymbol{x}_{\text{test}}) = k_{\boldsymbol{\theta}}(\boldsymbol{x}_{\text{test}}, \boldsymbol{x}_{\text{test}}) - \mathbf{k}_{\boldsymbol{\theta}}^{\mathsf{T}}(\boldsymbol{x}_{\text{test}})(\mathbf{K}_{\boldsymbol{\theta}} + \sigma^2 \mathbf{I})^{-1} \mathbf{k}_{\boldsymbol{\theta}}(\boldsymbol{x}_{\text{test}}).$$

The optimal kernel hyperparameters $\boldsymbol{\theta}^*$ can be learned by minimizing the negative log-likelihood [28].

## B.2 Learning Optimal Kernel Hyperparameters

To infer the optimal kernel hyperparameters $\boldsymbol{\theta}^*$ given $\mathcal{D}$, we can minimize the negative log marginal likelihood [28], leading to the following optimization problem:

$$\min_{\boldsymbol{\theta}} \mathcal{F}(\boldsymbol{\theta}) = \frac{1}{2}\det\left(\mathbf{K}_{\boldsymbol{\theta}} + \sigma^2\mathbf{I}\right) + \frac{1}{2}\mathbf{y}_{1:n}^{\mathsf{T}}(\mathbf{K}_{\boldsymbol{\theta}} + \sigma^2\mathbf{I})^{-1}\mathbf{y}_{1:n} + \text{cnst.} \tag{5}$$

Since the problem in the equation above is non-convex with respect to $\boldsymbol{\theta}$, typical solvers employ off-the-shelf packages that often utilize random restarts to escape local minima [42, 43, 79, 80].

## B.3 Acquisition Functions

We now provide additional information about the Probability of Improvement (PI) [61], the Lower Confidence Bound (LCB) [62] and Thompson Sampling (TS) acquisition functions, assuming that the aim is to minimize the black-box function value.

**Probability of Improvement:** The PI acquisition function is closely related to EI in that it also measures the utility of new query points with respect to the best black-box function value observed so far, $y^\star$. Contrary to EI, however, $\alpha^{(\mathrm{PI})}(\boldsymbol{x})$ judges the probability of acquiring new gains compared to $y^\star$ using the following definition:

$$\alpha^{(\mathrm{PI})}(\boldsymbol{x}) = \mathbb{E}_{f(\boldsymbol{x})|\mathcal{D}_{i-1},\boldsymbol{\theta}^\star}\left[\mathbb{I}\left[f(\boldsymbol{x}) < y^\star\right]\right],$$

where $\mathbb{I}$ is the indicator function that evaluates to 1 if $f(\boldsymbol{x}) < y^\star$ and to zero otherwise. Intuitively, the PI acquisition function estimates the probability of the black-box function value at $\boldsymbol{x}$ being lower than the lowest black-box function value observed so far.

**Lower Confidence Bounds (LCB):** Compared to the EI [59, 60] and the PI acquisition function, the LCB [62] trades-off the mean and the variance of the posterior distribution through an additional tuneable hyperparameter $\beta \in \mathbb{R}^+$:

$$\alpha_\beta^{(\mathrm{LCB})}(\boldsymbol{x}) = -\mu_{\boldsymbol{\theta}^\star}(\boldsymbol{x}) - \sqrt{\beta}\sigma_{\boldsymbol{\theta}^\star}(\boldsymbol{x}).$$

When minimizing a black-box function $f$, LCB estimates the potential minimum value at a given query location $\boldsymbol{x}$. Conversely, when maximizing the black-box, the Upper Confidence Bound (UCB) [62] should be used instead of the LCB, which is defined as:

$$\alpha_\beta^{(\mathrm{UCB})}(\boldsymbol{x}) = \mu_{\boldsymbol{\theta}^\star}(\boldsymbol{x}) + \sqrt{\beta}\sigma_{\boldsymbol{\theta}^\star}(\boldsymbol{x}).$$

**Thompson Sampling (TS):** The TS [63] acquisition function balances the exploration-exploitation trade-off by sampling function values from the posterior distribution of the black-box function and selecting the next query point based on the sampled values. In each iteration, TS samples potential query points $\boldsymbol{x}$ from the state space, estimates their mean black-box function values (see Appendix B.1), and selects the query point associated with the best-performing estimated mean value. In our proposed framework, the TS acquisition function is currently only utilized by BO methods relying on the Linear Regression surrogate model [1].

# Appendix C  Available Benchmarks

To enable the wider adoption and systematic evaluation of MCBO methods, our proposed framework addresses the need for standardized development and benchmarking methodologies. Building on the API for defining optimization problems, we provide a broad suite of real-world and synthetic benchmarks in our open-source software. These benchmarks consist of a diverse spectrum of optimization problems and their associated domains, covering a wide range of dimensionalities and difficulties. We provide the full list of the implemented tasks along with their domains in Table 2.

Table 2: Tasks implemented in the MCBO framework and the optimization domains they support. In this table, we assume that $x = [c, i, h]$ is partitioned into continuous variables $c$, integer-valued variables $i$ and categorical variables $h$. $\min_c$ and $\max_c$ specify arbitrary bounds for continuous variables. $\{\min_i, \max_i\}$ defines a set containing all integers between $\min_i$ and $\max_i$. $N_c$, $N_i$ and $N_h$ specify the number of continuous-valued, integer-valued and combinatorial variables respectively. $M_{N_i}$ specifies the number of categories the $i^{\text{th}}$ combinatorial variable can take.

| Objective $f$ | Inputs |
|---|---|
| SFU Test Functions | Arbitrary combination of input types and dimensionalities
$c \in [\min_c, \max_c]^{N_c}$
$i \in \{\min_i, \max_i\}^{N_i}$
$h \in \{h_1, ..., h_{M_1}\} \times ... \times \{h_{N_h}, ..., h_{M_{N_h}}\}$ |
| Ackley-20D (included in SFU) | 11 categories per variable
$h \in \{-32.768, -26.214, ...26.214, 32.768\}^{20}$ |
| Ackley-53D (included in SFU) | $c \in [-1, 1]^3$
$h \in \{0, 1\}^{50}$ |
| Pest Control | Pesticide choice at each stage (or use no pesticide)
$h = \{\text{No pesticide}, \text{Pest. 1}, \text{Pest. 2}, \text{Pest. 3}, \text{Pest. 4}\}^{25}$ |
| AIG Flow Tuning | $h \in \{\text{rewrite, rewrite -z, refactor, refactor -z, resub,}$
$\text{balance, \&blut, \&sopb, \&dsdb, fraig}\}^{20}$ |
| MIG Flow Tuning | $h \in \{\text{balance, cut rewrite, cut rewrite -z, refactor,}$
$\text{refactor -z, resubstitute, functional\_reduction}\}^{20}$ |
| Antibody Design | Let $AA$ be the set of all 20 naturally occurring amino acids
$h \in AA^{11}$ |
| RNA Inverse Folding | $h \in \{A, C, G, U\}^{N_h}$ |
| AIG Flow. and Hyp. Tuning | Sequence:
$\quad h \in \{\text{rewrite, rewrite -z, refactor, refactor -z, resub,}$
$\quad \text{balance, \&blut, \&sopb, \&dsdb, fraig}\}^{20}$
Hyperparameters of the operators:
$\quad$14 Boolean Variables and 22 integer variables |
| XGBoost - MNIST | $c_1$ is learning rate, $c_2$ is min split loss, $c_3$ is subsample,
$c_4$ is reg lambda, $i_1$ is max depth, $h_1$ is booster,
$h_2$ is grow policy and $h_3$ is objective.
$c_1 \in [10^{-5}, 1]$
$c_2 \in [0, 10]$
$c_3 \in [0.001, 1]$
$c_4 \in [0, 5]$
$i_1 \in \{1, 10\}$
$h_1 \in \{\text{gbtree, dart}\}$
$h_2 \in \{\text{depthwise, lossguide}\}$
$h_3 \in \{\text{multi:softmax, multi:softprob}\}$ |
| $\nu$-SVR - Slice | $c_1$ is $\epsilon$, $c_2$ is $C$, $c_3$ is $\gamma$ and $h$ concerns features
$c_1 \in [0.01, 1]$
$c_2 \in [0.01, 100]$
$c_3 \in [0.1, 10]$
$h \in \{\text{exclude, include}\}^{50}$ |

**Synthetic Tasks** Our suite of synthetic benchmarks provides a controlled setting for evaluating the performance of MCBO methods, where the true optima are often known, and objective function evaluations are inexpensive, enabling rigorous and scalable benchmarking. The benchmarks include the 21 Simon Fraser University (SFU) test functions described by [33], which are generalized to $d$-dimensional domains. These functions cover a range of optimization difficulties such as steep ridges, many local minima, valley-shape functions, and bowl-shaped functions. To increase their versatility, we have extended these functions to handle continuous, discrete, nominal, and ordinal variables, as well as combinations of these variable types. As part of the implementation of the SFU

test functions, we also include the Ackley -53D [5, 6] task, which is a special case of the Ackley function.

In addition, we include the pest control task used in [2, 5, 6], which involves optimizing the use of pesticides in a chain of 25 stations to minimize the number of products with pests while minimizing expenses on pesticide control. This task involves selecting from four different pesticides at each station, with varying prices and effectiveness, and has complex dynamics due to interactions between pesticides and pests. The pest control task provides a challenging optimization problem with high-order interactions.

**Real-World Tasks**    To evaluate the effectiveness of MCBO methods on real-world problems with practical objectives and constraints, our benchmark suite also includes a diverse set of real-world tasks spanning four domains: logic synthesis optimization, antibody design, RNA inverse folding, and hyperparameter tuning of ML models. We have selected these tasks for their relevance to different application domains and their challenging optimization landscapes.

**Logic Synthesis**: In the logic synthesis domain, we support benchmarks that involve optimizing the gate-level representation of Boolean circuits using a sequence of transformative operations. We consider two standard directed graph representations of Boolean circuits: the And-Inverter Graph (AIG) [71] and the Majority-Inverter Graph (MIG) [81]. We have a separate benchmark for each representation. Furthermore, we include an implementation of the AIG optimization benchmark that optimizes both the sequence of transformative operations and the hyperparameters of the underlying transformative algorithms. The AIG sequence and AIG sequence and hyperparameter optimization benchmark implementations rely on the ABC [71] codebase for applying transformative operations to the AIG representation, while the MIG sequence optimization benchmark relies on Mockturtle [82] library. All three logic synthesis benchmarks can be used to optimize any Boolean circuits, and notably the 20 Boolean functions from the open-source *EPFL Combinational Benchmark Suite* [83] that we use in our experiments.

**Antibody Design**: In the antibody design benchmark, we aim to optimize the CDRH3 sequence of an antibody for binding to a specific antigen. To achieve this, we use the Absolut! [84] software as an *in silico* framework for approximating the binding energy between an antibody and an antigen. Furthermore, we follow the recommendations of Khan et al. [72] for the developability constraint and check whether a CDRH3 sequence has no more than five consecutive amino acids that are identical, whether its net charge is in the interval $[-2, 2]$, and whether it is free of undesirable glycosylation motifs [85]. Our antibody design benchmark suite includes a diverse set of 159 antigen-CDRH3 binding tasks, where each task corresponds to a different antigen.

**RNA Inverse Folding:** In the RNA Inverse Folding task, the goal is to find one or more RNA sequences that fold into a target secondary structure. This task is of utmost importance to the RNA design process as it enables the design of novel RNA molecules with specific functions, such as catalysis and regulation [86–88]. As done in [73], we also on the ViennaRNA folding package [89] to model RNA inverse folding. However, as the ViennaRNA package does not deal with pseudoknots that can naturally occur in RNA structures, we further constrain the secondary structure by ensuring that it cannot have two base pairs $(i, j)$ and $(k, l)$ with $i < k < j < l$. In our benchmarking suite, we rely on the EteRNA100 dataset [90] to facilitate benchmarking on a collection of one hundred RNA secondary structures.

**Hyperparameter Optimization of Machine Learning Models:** Hyperparameters are critical tuning parameters that can greatly affect the performance of machine learning (ML) models. Optimizing these hyperparameters can improve the model's accuracy and robustness. In this work, we implement two tasks for hyperparameter optimization: one for the XGBoost [75] model on the MNIST [76] dataset (or any other dataset accessible through the Scikit-Learn [91] API), and the other for the $\nu$-Support Vector Regression ($\nu$-SVR) [77] model on the UCI slice dataset [78].

The search space of the XGBoost task comprises three categorical and five numerical variables. The categorical variables include the choice of the booster type, the grow policy, and the training objective. The numerical variables include the learning rate, max tree depth, minimum split loss, amount of regularization and the sub-sample magnitude.

For the $\nu$-SVR task, we use the Scikit-Learn [91] implementation of the $\nu$-SVR model. Similar to [6], we make the search space to include 3 continuous hyperparameters of the SVR, $C$, $\epsilon$, and $\gamma$,

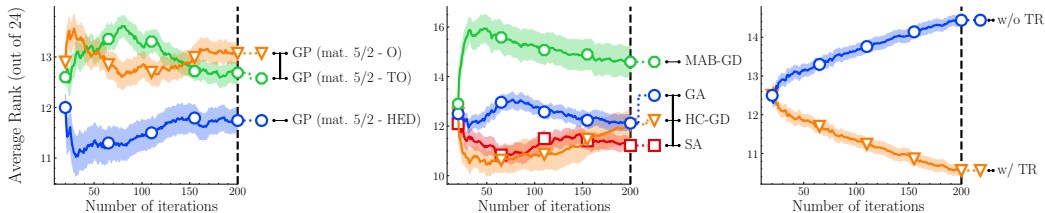

Figure 6: Evolution of the average rank of Mixed-variable BO algorithms across four tasks, aggregated by the surrogate model (left), acquisition optimizer (centre), and use of a trust region (right).

as well as 50 Boolean parameters corresponding to feature selection. Originally, the slice dataset comprises 384 features, but we reduce this number to 50 by fitting an XGBoost model and by keeping the 50 most important features according to the model. Then the 50 Boolean variables of the search space corresponds to the inclusion or exclusion of each of the retained features for the fit of a $\nu$-SVR model. The task consist in determine which features and values of the hyperparameters allow the minimization the RMSE obtained by the corresponding $\nu$-SVR model on a held-out test set.

## Appendix D  Benchmarking Implemented Mixed-Variable Algorithms

### D.1  Analyzing Mixed-variable BO Design Choices

**Surrogate model**  We observe in Fig. 6 (left) that BO methods using the GP (mat. 5/2 - HED) surrogate have a statistically significant lower rank compared to methods that use the GP (mat. 5/2 - O) and GP (mat. 5/2 - TO) surrogate. We hypothesize that this is because the HED kernel was better suited to modelling the interactions between the combinatorial variables for the four considered black-box functions.

**Acquisition function optimizers**  Fig. 6 (centre) reveals that the MAB-GD acquisition optimizer significantly underperformed on average compared to the remaining acquisition optimizers. It also shows that the HC-GD and SA acquisition optimizers are best suited for scenarios with low budgets, while there is no statistically significant difference between their performance and that of the GA acquisition optimizer at high budgets.

**Trust region (TR)**  Finally, Fig. 6 (right) encourages the the use of a TR in mixed-variable BO, as methods working with a dynamic TR to fit local models and constraining acquisition optimization provide consistently better suggestions compared to global approaches. This confirms and extends the findings of Wan et al. [5] and is consistent with our results presented in section 5.3.

### D.2  Which is the Most Robust Mixed-Variable Algorithm

We now consider each mix-and-match mixed-variable BO algorithm individually (without aggregating ranks by primitives) and compare them to non-BO baselines. We first order all the algorithms based on their average ranks across fifteen seeds and four tasks, and we show on Fig. 7 the performance of the three known BO solvers, the four non-BO baselines and the 2 (out of 21) new mix-and-match methods achieving the lowest rank.

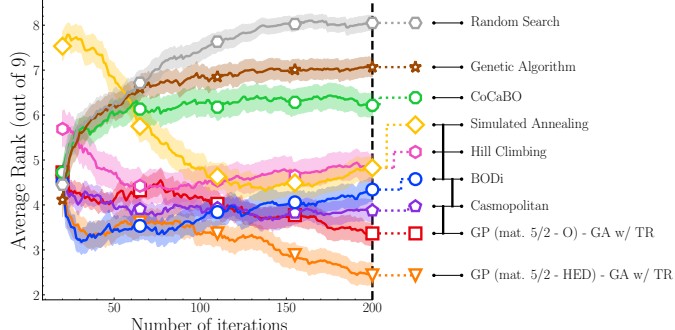

Figure 7: Average rank of known mixed-variable algorithms and the two best mix-and-match BO algorithms.

We observe that the new mixed-variable algorithm, GP (mat 5/2 - HED) - GA w/ TR, achieves a statistically significant average

rank improvement compared to the considered known mixed-variable BO algorithms and non-BO baselines. Notably, this algorithm is novel, and its design is aligned with our conclusions from Appendix D.1.

For each BO optimizer and the three best non-BO baselines, we show on Fig. 8 the evolution of the best objective values attained. We note that on three of the considered tasks (Ackley-53D, AIG Flow. and Hyp. Tuning and XGBoost - MNIST), a BO algorithm outperformed the considered non-BO solvers. However, on the final task, $\nu$-SVR - Slice, the HC and SA non-BO baselines attained comparable performance to the best-performing BO solvers, highlighting the need for further development of mixed-variable BO techniques when it comes to solving real-world tasks. Nevertheless, BO solvers are still the best on most tasks, though we observe that no single optimizer achieves the lowest objective value across all tasks.

## Appendix E   Notes on Compatibility of Implemented MCBO Primitives

The MCBO framework includes three core BO primitives: surrogate models, acquisition functions, and acquisition optimizers. Below we discuss some of the limitations on the mix-and-match compatibilities of the currently implemented MCBO primitives and their compatibility with various optimization domains.

### E.1   Surrogate Models

Our implementation features the following pre-implemented surrogate models:

- GP with SSK [4, 7]
- GP with overlap kernel (O) [3]
- GP with transformed-overlap kernel (TO) [5]
- GP with diffusion kernel (Diff.) [2]
- GP with dictionary-based kernel (HED) [6]
- GP with mixture kernel [5]
- Linear Regression [50] with the Horseshoe prior [67] using maximum likelihood, maximum a posteriori, and Bayes estimation (LHS) [1].

Below we provide some additional details about the compatibility of each of these models.

*GP (SSK)* is only applicable to combinatorial problems where each of the categorical variables shares the same possible categories.

*GP (O)* is only applicable to combinatorial problems.

*GP (TO)* is only applicable to combinatorial problems.

*GP (Diff.)* is only applicable to combinatorial problems. Also, as the GP (Diff.) model as proposed by Oh et al. [2] is really an ensemble of 10 models, any acquisition function the user wishes to combine with this model must be used to initialize the `SingleObjAcqExpectation` class, which can be subsequently mixed-and-matched with the model.

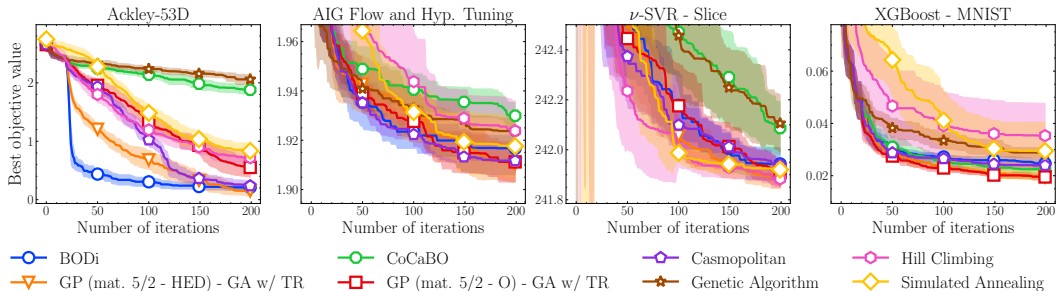

Figure 8: Evolution of the best observed value on each mixed-variable task achieved by the known mixed-variable optimizers and the two best mix-and-match BO solvers built with MCBO framework. In all tasks, the aim is to minimize the black-box. Hence, the lower the best observed value the better.

*GP (HED)* is only applicable to combinatorial problems.

*GP (mixture kernel)* is compatible with mixed-variable problems. The mixture kernel currently supports the use of the RBF or Mat. 5/2 kernel for numeric variables, and the use of the O, TO or HED kernel for categorical variables, resulting in six different potential GP (mixture kernel) models.

*LR (HS)* is only applicable to combinatorial problems.

## E.2 Acquisition Functions

Our implementation features the following pre-implemented acquisition functions:

- Expected Improvement (EI) [59, 60]
- Probability of Improvement (PI)[61]
- Lower Confidence Bound (LCB) [62]
- Thompson Sampling (TS) [63]

The *EI, PI* and *LCB* acquisition functions support all surrogate models except for the LR (HS) surrogate model. The reason for this is that the LR (HS) does not maintain an analytical estimate of the posterior but instead uses Gibbs sampling [1] to sample from it, making it impossible to calculate EI, PI and LCB in closed form. In contrast, the *TS* acquisition function only supports the LR (HS) surrogate model.

## E.3 Acquisition Optimizers

Our implementation features the following pre-implemented acquisition optimizers:

- Exhaustive Local Search (LS) [2]
- Genetic Algorithm (GA) [7]
- Simulated Annealing (SA) [1]
- Interleaved Search (Hill Climbing + Gradient Descent) (IS (HC + GD))[5]
- Interleaved Search (Multi-Armed Bandit + Gradient Descent) (IS (MAB + GD))[3]

Below we provide some additional details about the compatibility of each of these optimizers.

*LS* is only applicable to combinatorial problems.

*GA* is compatible with combinatorial and mixed-variable problems.

*SA* is compatible with combinatorial and mixed-variable problems.

*IS (HC + GD)* is compatible with combinatorial and mixed-variable problems. For combinatorial problems, this acquisition optimizer will simply perform hill climbing, while for purely numeric problems, it will perform gradient descent.

*IS (MAB + GD)* is compatible with combinatorial and mixed-variable problems. For combinatorial problems, this acquisition optimizer will simply use a MAB to optimize the combinatorial variables, while for purely numeric problems, it will perform gradient descent.

We also note that we have generalized the implementation of these acquisition optimizers so that **all of them support TR-constrained acquisition optimization**, extending [5, 68], and make them **handle simple input domain constraints** via rejection sampling and projections.

## E.4 Implemented Mix-and-Match Optimizers

Given the compatibilities outlined above, in the MCBO framework, for combinatorial formulations, we support 48 different mix-and-match BO optimizers. This results from combining the six different surrogate models suitable for combinatorial formulations (GP (SSK), GP (O), GP (TO), GP (Diff.), GP (HED) and LR), with the four compatible acquisition optimizers (LS, GA, SA and HC) both with and without a trust region constraint.

Similarly, for mixed-variable formulations, we can instantiate 24 different mix-and-match BO optimizers. This results from combining three compatible surrogate models (GP (Mat. 5/2 + O), GP (Mat. 5/2 + TO) and GP (Mat. 5/2 + HED)) with the four compatible acquisition optimizers (GA, SA, IS (HC + GD) and IS (MAB + GD)), both with and without a trust region constraint.

We note that 7 of these optimizers are known combinatorial and mixed-variable BO algorithms. After accounting for the fact that some mix-and-match combinatorial optimizers are special cases of the known BO algorithms, this results in a total of 47 novel mix-and-match BO algorithms.

## Appendix F  Experimental setup

### F.1  Hyperparameters

We share in Table 3 a comprehensive list of the hyperparameters used during training and inference in all experiments. More details can be found in the associated code repository.

Table 3: List of MCBO primitives hyperparameters.

| Surrogate model | |
|---|---|
| GP mixed-kernel form | $\lambda(K_{\text{cat}} + K_{\text{num}}) + (1 - \lambda)(K_{\text{cat}} \cdot K_{\text{num}})$ |
| GP Kernel for numeric variables | Matértn-5/2 |
| GP noise level lower bound | $10^{-5}$ |
| GP fit negative log-likelihood optimizer | Adam(lr $= 0.03$) [92] for 100 epochs |
| Diffusion GP characteristics [2] | n_models $= 10$, n_burn_init $= 100$ |
| Bayesian linear regression SH estimator [1] | Order $= 2$, Threshold $= 0.1$, a $= 2$, b $= 1$, n_gibbs $= 10^{-3}$ |
| **Acquisition function optimizer** | |
| LS | n_random_vertices=20000, n_greedy_ascent_init=20, n_spray=10 |
| SA | num_iter=100, n_restarts=3, init_temp=1, tolerance=100 |
| GA | ga_num_iter=500, ga_pop_size=100, |
| | cat_ga_num_parents=20, cat_ga_num_elite=10 |
| IS (HC + GD) | n_iter=100, n_restarts=3, max_n_perturb_num=20, |
| | num_optimizer=Adam, num_lr=1e-3, nom_tol=100 |
| IS (MAB + GD) | max_n_iter=200, mab_resample_tol=500, n_cand=5000, |
| | n_restarts=5, num_optimizer=SGD, cont_lr=3e-3, cont_n_iter=100 |
| **Trust region Manager** | |
| tr_restart_acq_name | LCB |
| tr_restart_n_cand | min($100 \times$ num_dims, 5000) |
| tr_min_num_radius | $2^{-5}$ |
| tr_max_num_radius | 1 |
| tr_init_num_radius | $0.8\times$ tr_max_num_radius |
| tr_min_nominal_radius | 1 |
| tr_max_nominal_radius | num_nominal |
| tr_init_nominal_radius | round($0.8\times$ tr_max_nominal_radius) |
| tr_radius_multiplier | 1.5 |
| tr_succ_tol | 3 |
| tr_fail_tol | 40 |

## Appendix G  Implementation Details and Additional Features

### G.1  Input and Output Normalization

For stable learning with GPs, we normalize all inputs to the range $[0, 1]$ and standardize all black-box function values using the mean and standard deviation of all previously observed values. We apply this standardization independently during every iteration of BO, prior to fitting a model.

### G.2  Input Constraints

The MCBO framework supports cheap-to-compute constraints on the search space via rejection sampling. When defining an optimization problem via the `TaskBase` class, one can additionally define a list of functions which can each take a sample as input and returns whether the sample is valid or not. Then, we pass this list of functions as an argument when constructing a BO algorithm with the `BoBuilder` class, as we do for the Antibody design task to restrict the search space to feasible antibody sequences.

## G.3 Trust-Region-Based Acquisition Optimization

All implemented acquisition optimizers support trust-region-constrained acquisition optimization. Following [5], the trust region centred around the best input found so far in the current trust region, $\boldsymbol{x}^{\text{TR}}$, is defined as

$$\text{TR}(\boldsymbol{x}^{\text{TR}}) = \{\boldsymbol{x} \in \mathcal{X} \text{ s.t. } d_c(\boldsymbol{x}_c^{\text{TR}}, \boldsymbol{x}_c) \leq L_c \text{ and } d_{L_n}(\boldsymbol{x}_n^{\text{TR}}, \boldsymbol{x}_n) \leq 1\},$$

where $\boldsymbol{x}_c^{\text{TR}}$ and $\boldsymbol{x}_c$ are vectors containing all the combinatorial inputs in $\boldsymbol{x}^{\text{TR}}$ and $\boldsymbol{x}$ respectively, $d_c(\cdot, \cdot)$ is the Hamming distance, and $L_c$ is the size of the TR for combinatorial variables. Similarly, $\boldsymbol{x}_n^{\text{TR}}$ and $\boldsymbol{x}_n$ are vectors containing all the numeric and discrete variables in $\boldsymbol{x}^{\text{TR}}$ and $\boldsymbol{x}$ respectively, $d_{L_n}(\cdot, \cdot)$ is the maximum of the component-wise distance for numeric and discrete variables normalized by dimensional length scales $L_n \in \mathbb{R}_+^{\dim(\boldsymbol{x}_n)}$, i.e.

$$d_{L_n}(\boldsymbol{x}_n^{\text{TR}}, \boldsymbol{x}_n) = \max_{i \in \dim(\boldsymbol{x}_n)} \frac{|\boldsymbol{x}_n^{\text{TR}}[i] - \boldsymbol{x}_n[i]|}{L_n[i]}.$$

The radius of the combinatorial trust region, $L_c$, and the radii for the numeric variables, $L_n$, are adjusted dynamically during optimization. They are expanded on successive successes (i.e. when the best function value observed improves) and shrunk otherwise. They also have a predetermined maximum and minimum value. If any of the radii were to be expanded past their maximum allowable value, they would be capped at this value instead. However, if any of the radii were to be shrunk smaller than their minimum value, this would trigger a trust region restart.

During a trust region restart, all the radii are set to their initial value, and a global auxiliary surrogate model is used to determine a centre of a new trust region. Suppose an algorithm restarts its trust region for the $i^{\text{th}}$ time. First, the global surrogate model is fitted to a subset of data $D^* = \{\boldsymbol{x}_j^{\text{TR}}, y_j^{\text{TR}}\}_{i=1}^{j-1}$, where $\boldsymbol{x}_j^{\text{TR}}$ is the local maxima found after the $j^{\text{th}}$ restart, and $y_j^{\text{TR}}$ is its corresponding black-box function value. This auxiliary surrogate model is then used to define an acquisition function, which is then subsequently optimized to suggest a new trust region centre.

## G.4 Batch sampling

All currently implemented acquisition function optimizers support batch sampling using the Kriging Believer strategy [93] to select $b$ points sequentially. To sample the $i^{\text{th}}$ point, the Kriging Believer strategy replaces the black-box function evaluation for the $(i-1)^{\text{th}}$ point with the GP prediction, *hallucinating* what the black-box function value could have been, and retrains the GP on the aggregated dataset, before optimizing the acquisition function to obtain the $i^{\text{th}}$ suggestion.

