# OpenReview forum: "Framework and Benchmarks for Combinatorial and Mixed-variable Bayesian Optimization"
_NeurIPS.cc/2023/Track/Datasets_and_Benchmarks — NeurIPS 2023 Datasets and Benchmarks Poster_

### Official Review · Reviewer_X2sz · 2023-07-07
**Flexible Framework for Benchmarking and Evaluation of Bayesian Optimizers**

**Rating:** 7
**Confidence:** 3
**Clarity:** The paper is well-written.

**Strengths:**

Having a centralized repository of benchmark datasets will facilitate evaluation of new Bayesian optimizers and comparison with existing techniques. This reviewer is not aware of any similar public dataset.

The Python code for easily generating a new optimizer or to quickly implement existing optimizer also makes it much easier for researchers to compare other techniques. For a given problem, it may be possible to select a dataset that best matches the application, allowing an evaluation of different optimization techniques to see which performs best for this problem.

The research solution appears to be of high quality and the paper is well-written.

**Additional Feedback:**

On line 202, "suit" should be "suite".

**Correctness:**

The paper appears to be correct. The dataset is constructed properly. The evaluations that are presented are adequate to represent the contribution.

**Documentation:**

Yes. There is sufficient detail for both the dataset and the benchmarks.

**Ethics:**

No. This reviewer did not find anything that raised ethical concerns.

**Limitations:**

These issues are not discussed in the paper or the appendices. Other than what is mentioned in the previous section of this review, the breadth of coverage of the datasets and the process, if any, for enhancing the datasets with additional problem data should be addressed. In particular, a researcher may wish to use the Python script to evaluate different MCBO algorithms for their own dataset/problem.

**Opportunities For Improvement:**

The paper could be improved by providing more details on the use and limitations of the Python code to create new MCBO algorithms. Is it truly the case that any reasonable MCBO algorithm can be implemented just by setting a few configuration parameters? Are there some dimensions of algorithm design that aren't covered?

The paper does not provide the required checklist, so it's not possible to evaluate any potential ethical or societal implications. Intuitively, there don't seem to be any or many, but there may be some non-obvious issues.

**Relation To Prior Work:**

As noted above, the work is novel in providing both datasets for MCBO algorithms and also a Python code for significantly easing the task of comparing a novel algorithm with existing research. The related work does not address these points as thoroughly as this paper does.

**Summary And Contributions:**

This paper offers a framework for benchmarking of Mixed-variable and combinatorial Bayesian optimizers. The first primary contribution of the paper is a set of benchmark datasets drawn from both real-world applications and synthetic datasets to allow a comprehensive evaluation of the various methods. The second primary contribution is a Python code that greatly simplifies the process of implementing and testing new or existing Bayesian optimizers.

---

> ### Author Response · Authors · 2023-08-24
> **Rebuttal by Authors**
>
> We thank the reviewer for their time and for stressing the quality of the writing. We try to answer the questions regarding the limitations of our framework below.
>
> > The paper could be improved by providing more details on the use and limitations of the Python code to create new MCBO algorithms. Is it truly the case that any reasonable MCBO algorithm can be implemented just by setting a few configuration parameters?
>
> We would like to highlight that there are typically two cases to consider:
> 1. Implementing a "new" MCBO algorithm that is part of the ones presented as "New" in Figure 1 is really as simple as specifying the name of the surrogate model, the acquisition function optimizer, the acquisition function, and the trust-region policy to instantiate a `BoBuilder`. Additionally, it is possible to pass module-specific kwargs to alter the default parameters of the selected modules (e.g. changing the noise prior of a Gaussian Process, changing the beta factor of the Lower-Confidence-Bound acquisition function, etc.)
> 2. Implementing a "new" MCBO algorithm that utilises a BO primitive not already available within our library. This requires a bit of work to implement the primitive and to make it available to the BoBuilder class that allows for mixing and matching of BO components.
> To ease this integration, we added [step-by-step instructions](https://github.com/huawei-noah/HEBO/tree/master/MCBO/tutorials#adding-new-bo-modules) in our codebase, showing where the new primitives should be added, and pointing to examples to inspire from.
> We can mention that the integration of new BO primitives has already happened with the inclusion of a new surrogate model (a GP based on a random-tree-decomposition kernel - GPRD), a new acquisition function (additive LCB (A-LCB)) and a new acquisition function optimizer (message-passing (MP)) [1]. All of these modules have been straightforwardly added to the BoBuilder to be instantiated through our unified API.
> Although on the one hand, MP and A-LCB components are hardly amenable to mixing-and-matching as they only operate with a surrogate model having special structural properties, on the other hand, this surrogate model can be associated with any acquisition function optimizer and acquisition function, and can be used with a trust region simply, through the Bobuilder.
>
> > Are there some dimensions of algorithm design that aren't covered?
>
> There are indeed dimensions of extended MCBO designs that are not covered and some of them are already under development. Extensions would include the ability to handle multi-objective black-box functions, supporting optimization under black-box constraints (we have already included a constrained-Expeected Improvement but still need to modify the BoBase to take as input optional surrogate models to fit the black-box constraints), or the ability to interact with a multi-fidelity black-box. All these variants of the core BO algorithm require additional refactoring beyond adding a simple module, but providing a very modular framework for the standard BO scenario is a fundamental step in the development of an even more ubiquitous library.
>
> [1] Ziomek, J., Bou-Ammar, H., (2023).  Are Random Decompositions all we need in High Dimensional Bayesian Optimisation?, Proceedings of the 40th International Conference on Machine Learning, PMLR 202:43347-43368

---

> > ### Comment · Reviewer_X2sz · 2023-08-28
> > **Reply**
> >
> > These comments have addressed my comments. I'm satisfied with these revisions/comments.

---

### Official Review · Reviewer_fKag · 2023-07-21
**Review - Framework and Benchmarks for Combinatorial and Mixed-variable Bayesian Optimization**

**Rating:** 7
**Confidence:** 3
**Correctness:** The claims are correct to the best of…

**Strengths:**

The paper is written very well and accessible and the provided code is written very well too (I evaluated only samples though). The representation of existing MCBO algorithms according to their primitives and the resulting matrix of yet unseen algorithms (Fig. 1) is very nice. Equally enlightening is Figure 2, about the performance of the individual MCBO primitives.

As stated in the paper, the presented framework can help in evaluating individual yet to be developed primitives in a standardized way. Also the extension to non MCBO algorithms to broaden the comparison is very helpful for this endeavor.

**Additional Feedback:**

-

**Clarity:**

The paper is written very well. It guides also the non familiar reader through the whole paper and defers implementation details to the appendix.

**Documentation:**

The provided documentation and code are very well.

**Ethics:**

No ethics concern visible.

**Limitations:**

-

**Opportunities For Improvement:**

-

**Relation To Prior Work:**

The paper discusses clearly what differentiates it from previous contributions, namely:
- the focus on BO of combinatorial and mixed-variable spaces
- the ease of constructing new BO algorithms based on mixing of already implemented BO primitives

**Summary And Contributions:**

The paper presents a python framework for the efficient comparison of Mixed-variable and Combinatorial Bayesian Optimization (MCBO) algorithms. To facilitate the comparison the framework allows the simple combination of Bayesian Optimization (BO) primitives.  This allows the fast creation and evaluation of yet undocumented MBCO algorithms. For the demonstration of the versatility of the framework an insightful experimented is conducted and evaluated.

I believe a higher rating was appropriate, but I am not really familiar with the subject, so this has to be evaluated by someone more familiar.

---

> ### Author Response · Authors · 2023-08-24
> **Rebuttal by Authors**
>
> We thank the reviewer for their time and for having tested our codebase. We appreciate the assesment that our paper is also accessible to readers that are not experts on BO as we have aimed to target both BO researchers looking for testing their new methods, but also non-experts who have real-scenario  mixed-variable or combinatorial black-box functions to optimize. To ease the adoption of our framework by both groups, we have extended the documentation of the library and provided [tutorials](https://github.com/huawei-noah/HEBO/tree/master/MCBO/tutorials) that describe step-by-step instructions on how to run optimization of a custom black-box, or how to include a new method and run it on our benchmarking tasks.

---

### Official Review · Reviewer_Vgsh · 2023-07-21
**Solid framework for combinatorial and mixed-variable BO**

**Rating:** 7
**Confidence:** 4
**Clarity:** The paper is written well.

**Strengths:**

Combinatorial and especially mixed-variable BO is timely research and especially in the context of black box functions, these kinds of problems appear to gain increased interest.
Relevance and significance is given.
The framework itself appears to be well designed and user friendly.

**Additional Feedback:**

As far as I am aware, the framework currently provides little utility regarding the analysis of runs of an algorithm on a benchmark.
It would be nice or an additional plus to include such (basic) utility (standard anytime plots, maybe expected running time analysis in the case of synthetic benchmark functions for which the optimum is known etc.)

As a minor question: Is it also possible to use a simple random search as a baseline for acquisition function optimization? I would expect it to get out-performed by the other acquisition function optimizers but supporting it would allow for such a straightforward comparison.

Also, would it be possible to support other sampling strategies to generate the initial design (sobol or lhs instead of uniform sampling)?

**Correctness:**

The framework and benchmarks appear to be constructed in a sound way. The empirical protocol is sensible. Regarding the statistical analysis, please see my comments above regarding the use of an overall test.

**Documentation:**

Documentation appears to be sufficient although a maintenance plan is missing. Benchmarks appear to be reproducible.

**Ethics:**

No ethical concerns are suspected.

**Limitations:**

As the authors propose a framework for implementing and benchmarking combinatorial and mixed-variable BO algorithms, direct societal impact is difficult to discuss.

**Opportunities For Improvement:**

Besides of the modular inclusion of the TR approach, the framework is more or less a standard modular BO framework.

Regarding the review and discussion of existing benchmark functions or suites, I believe the paper currently falls somewhat short:

1) The BBOB functions as part of the COCO suite provides a mixed-integer benchmark [1] that has been adopted in the black box optimization community.

2) Hyperparameter optimization (HPO) benchmarking suites such as HPOBench [2], HPO-B [3], or YAHPO-Gym [4] all provide mixed-integer or mixed-variable benchmark problems that have been adopted in the HPO community.

The authors should discuss existing benchmarking suites and illustrate how their own functions differ or go beyond existing ones in more detail.

Regarding acquisition functions, it would have been nice to also implement acquisition functions such as Knowledge Gradient or entropy based ones (ES, PES, MES, or JES) which on the one hand are much more difficult to implement (especially for more inexperienced users and therefore it would be nice to have solid open source implementations users can rely o) but have been shown to sometimes outperform classical acquisition functions such as EI, PI, LCB, or TS.

Moreover, I would appreciate if the benchmark functions could be explained in more detail in the appendix (search space, domain, codomain).
Table 1 in the appendix summarizes the functions nicely but still lacks detailed information.

Currently, the choice of 200 function evaluations per benchmark function (regardless of the nature and dimensionality of the function) seems somewhat arbitrary (similarly the fixed initial design size of 20 points). Can the authors provide some insight into this choice?

In the current experimental procedure, multiple paired Wilcoxon signed rank tests are carried out.
It would be preferable to first conduct an overall test such as the Friedman test and only conduct (for multiple testing corrected follow up tests) if the global null hypothesis of no difference between any pair of algorithms can be rejected.

Moreover, I am not sure whether the question of "Which is the Best Performing Combinatorial algorithm?" is a sensible one. Given enough and diverse and structurally different functions, I would expect that compensatory effects should appear (some algorithms performing strong on some functions, other performing strong on others) as there cannot be a single algorithm that solves all problems best.
It might be interesting to conduct a statistical analysis to find patterns, i.e., for which kind of problems do which algorithms perform well.

[1] https://dl.acm.org/doi/10.1145/3321707.3321868
[2] https://arxiv.org/abs/2109.06716
[3] https://arxiv.org/abs/2106.06257
[4] https://arxiv.org/abs/2109.03670

**Relation To Prior Work:**

Prior work is discussed sufficiently with the exception of the discussion of existing benchmarking suites in the field of black box optimization and HPO.

**Summary And Contributions:**

The paper proposes a framework to implement combinatorial and mixed-variable Bayesian Optimization (BO) algorithms and benchmarks existing algorithms and baselines as well as novel algorithms constructed based on the modular framework on a set of benchmark problems.
The framework itself is built modular based on the primitives of a surrogate model, an acquisition function and an acquisition function optimizer (as in standard BO) which can operate on a trust region (somewhat non-standard but this can be central for good performance of combinatorial and mixed-variable BO).
Somewhat interesting findings regarding the choice of surrogate model (especially GP kernel) as well as acquisition function optimizer (i.e., local search vs. GA) and trust region mechanisms are presented and discussed.

---

> ### Author Response · Authors · 2023-08-24
>
> We thank the reviewer for appreciating the significance of the problem we tackle and for underlining the clarity of the writing. We thank them for raising interesting points that we discuss below.
>
> > The authors should discuss existing benchmarking suites and illustrate how their own functions differ or go beyond existing ones in more detail.
>
> We appreciate the reviewer's feedback concerning the comparison of our benchmark functions with existing benchmark suites. It is important to note that the BBOB functions integrated into the COCO suite consist of 24 synthetic mixed-variable functions with fixed dimensionalities. In our framework, we have extended some of these functions, including the sphere, rotated hyper ellipsoid, Rastrigin, and Rosenbrock test functions, to accommodate arbitrary dimensional domains and combinations of variable types. In addition to this, the SFU test functions that are a part of our benchmarking suite also include additional synthetics test functions that follow the same generalization principles.
>
> While acknowledging the broader range of benchmarks offered by HPOBench, HPO-B, and YAHPO-Gym, it's crucial to highlight a key distinction. These suites predominantly centre around hyperparameter optimization, whereas our framework boasts a more expansive coverage of problem domains. Our implemented tasks encompass a diverse array of areas such as hyperparameter tuning, antibody design, Electronic Design Automation, RNA inverse folding, and synthetic tasks. By embracing this multifaceted approach, we believe our benchmark suite can be used to yield more generalisable and robust conclusions compared to scenarios that hinge solely on a single family of tasks.
>
> Taking advantage of the extra page, we have now updated our related work section to distinguish our benchmarks from existing benchmarking suites.
>
> > Regarding acquisition functions, it would have been nice to also implement acquisition functions such as Knowledge Gradient or entropy based ones ...
>
>  This is a very relevant remark, and we want to acknowledge that supporting non-myopic acquisition functions for combinatorial and mixed BO is an interesting direction, as well as a challenging one.
>  To the best of our knowledge, even for advanced and approximate versions of non-myopic acquisition functions such as One-Shot Knowledge Gradient (KG) [1], Predictive Entropy Search (PES) [2] and Max-value Entropy Search (MES) [3], no work demonstrated the feasibility of efficiently operating BO equipped with such acquisition function when dealing with mixed or combinatorial black-box functions with more than 10 dimensions. For example, GIBBON [4] optimizes at most an 8-D black-box. Note that when GIBBON runs on SMILES optimization, the search space is actually limited to a pre-defined set of molecules and does not optimize the acquisition function over the all search-space, which does not fully reflect the difficulty of the high-dimension combinatorial optimization that needs to be solved). Similarly, the PES-based method by [5] presents results for up to 10-dimensional search spaces.
>  As our benchmark was designed to contain reasonably high-dimensional (>20D) real-world mixed and combinatorial tasks, we decided to start by focusing on providing a framework supporting the widely adopted myopic acquisition functions, and leave the integration or development of non-myopic acquisition functions (along with the optimization techniques amenable to mixed search space) for a future development.
>
> > Moreover, I would appreciate if the benchmark functions could be explained in more detail in the appendix
>
> As a complement to the paragraphs dedicated to the presentation of each benchmark in Appendix C, we have now updated the table detailing the implemented benchmarks in our appendix to clarify the dimensionality and the nature of each variable in their search spaces.

---

> > ### Author Response · Authors · 2023-08-24
> >
> > > Currently, the choice of 200 function evaluations per benchmark function (regardless of the nature and dimensionality of the function) seems somewhat arbitrary (similarly the fixed initial design size of 20 points). Can the authors provide some insight into this choice?
> >
> > The decision to utilize an initial design size of 20 points, regardless of problem dimensionality, aligns with established practices observed in prior research, including [6, 7, 10, and 9] which all adopted an initial design size of 20 points for all their considered tasks irrespective of their dimensionality. This approach also closely resembles the design size of 24 points employed by [8] for all considered problems.
> >
> > Regarding the choice of 200 function evaluations per benchmark function, our rationale revolves around balancing computational efficiency while accommodating even the more resource-intensive methods, such as [7] and [6], within reasonable wall-clock time constraints. The selected count of 200 evaluations mirrors the choices made by [9, 10, 11], and falls within the range of black-box function evaluations adopted by other benchmarks like [10] (150-400), [7] (170-270), and [6] (100-250).
> >
> > > In the current experimental procedure, multiple paired Wilcoxon signed rank tests are carried out. It would be preferable to first conduct an overall test such as the Friedman test and only conduct (for multiple testing corrected follow up tests) if the global null hypothesis of no difference between any pair of algorithms can be rejected.
> >
> > Indeed, we duly conducted a Friedman test before carrying out the post-hoc Wilcoxon test whenever we compared more than two elements. For all the provided results, the null hypothesis that all the compared elements perform equally was always rejected. In the revised version of our paper, we now mention that we first conducted a Friedman test before doing the post-hoc analysis.
> >
> > > Moreover, I am not sure whether the question of "Which is the Best Performing Combinatorial algorithm?" is a sensible one ...
> >
> > We appreciate the reviewer's insightful comment regarding the appropriateness of the question "Which is the Best Performing Combinatorial algorithm?" in our work. We agree that the original phrasing did not effectively convey our intended focus on diversity and robustness. In response, we have revised our experiments section to more accurately reflect our goal, which is now framed as "Which is the Most Robust Combinatorial algorithm?" By "robust," we mean an algorithm that demonstrates consistent and strong performance across a broad spectrum of optimization problems from different domains characterized by varying complexities,
> > and which could be the default choice if one needs to optimize a new black-box on without particular prior knowledge.
> >
> > In line with the reviewer's comment, we have indeed observed that various algorithms exhibited varying performance across the different considered functions. This observation holds true not only for BO algorithms but also for the non-BO baselines, which were found to perform best for three out of the ten considered tasks. The individual results for all considered tasks are shown in Figures 4 and 8, highlighting the nuanced performance variations. While our paper currently focuses on providing insights into robustness, we concur with the reviewer's suggestion that conducting a statistical analysis to discern patterns among algorithmic performances on different problem types is a compelling avenue for future research.

---

> > > ### Author Response · Authors · 2023-08-24
> > >
> > > > Documentation appears to be sufficient although a maintenance plan is missing.
> > >
> > > Based on this comment, we have now included our [maintenance plan](https://github.com/huawei-noah/HEBO/tree/master/MCBO#library-roadmap) for MCBO library. We would also like to note that we have now further improved our documentation and provide additional [tutorials](https://github.com/huawei-noah/HEBO/tree/master/MCBO/tutorials) on optimizing a custom black-box, adding new BO modules, evaluating a new BO algorithm on the MCBO tasks and estimating the runtime of an algorithm.
> > >
> > > > As far as I am aware, the framework currently provides little utility regarding the analysis of runs of an algorithm on a benchmark. It would be nice or an additional plus to include such (basic) utility (standard anytime plots, maybe expected running time analysis in the case of synthetic benchmark functions for which the optimum is known etc.)
> > >
> > > We thank the reviewer for their suggestion. We now provide a function for estimating the runtime of an algorithm on an arbitrary benchmark (including custom ones), and a [tutorial](https://github.com/huawei-noah/HEBO/tree/master/MCBO/tutorials#runtime-estimation) illustrating the use of this function. Furthermore, we also provide functions for visualising the regret trajectory of optimizers, including the results presented in our paper. The usage of this function is illustrated in this [jupyter notebook](https://github.com/huawei-noah/HEBO/blob/master/MCBO/tutorials/tuto_results_viz.ipynb).
> > >
> > > > Is it also possible to use a simple random search as a baseline for acquisition function optimization?
> > >
> > > Indeed this is possible. In fact, we have now implemented a [random-search-based acquisition optimizer](https://github.com/huawei-noah/HEBO/blob/master/MCBO/mcbo/acq_optimizers/random_search_acq_optimizer.py), and have fully integrated it into the BoBuilder class, enabling the construction of MCBO algorithms utilising this acquisition optimizer with a single line of code.
> > >
> > > > Also, would it be possible to support other sampling strategies to generate the initial design (sobol or lhs instead of uniform sampling)?
> > >
> > > Following this suggestion, our `BoBuilder` can now take as input an `init_sampling_strategy`  to generate potentially better initial designs than the ones from default uniform sampling.
> > > For now, we support Sobol sampling with or without scrambling, as demonstrated in the example of the [MCBO README.md](https://github.com/huawei-noah/HEBO/tree/mcbo/MCBO#simple-optimization-example). We first draw samples from the Sobol engine with the same dimension as the search space, and we convert the samples in $[0, 1]^d$ to samples in the original space by rescaling the samples and calling the search space `inverse_transform` method (that will project continuous values to valid parameter value).
> > > Nonetheless, we note that even though Sobol sampling provides good space-filling guarantees in the continuous setting, they do not hold when the samples are discretized. Therefore, we also plan to explore alternative strategies such as sampling from k-Determinantal Point Processes [12].

---

> > > > ### Author Response · Authors · 2023-08-24
> > > >
> > > > References:
> > > >
> > > > [1]  J. Wu and P. Frazier. The parallel knowledge gradient method for batch bayesian optimization. NeurIPS 2016.
> > > >
> > > > [2] Hernandez-Lobato, J.M., Gelbart, M., Hoffman, M., Adams, R. P.; Ghahramani, Z.. (2015). Predictive Entropy Search for Bayesian Optimization with Unknown Constraints. Proceedings of the 32nd International Conference on Machine Learning</i>, in <i>Proceedings of Machine Learning Research, 37:1699-1707
> > > >
> > > > [3] Wang Zi, Jegelka S., Max-value Entropy Search for Efficient Bayesian Optimization. Proceedings of the 34 th International Conference on Machine Learning, Sydney, Australia, PMLR 70, 2017.
> > > >
> > > > [4] Henry B. Moss, David S. Leslie, Javier Gonzalez, Paul Rayson:
> > > > GIBBON: General-purpose Information-Based Bayesian Optimisation. Journal of Machine Learning Research 22: 235:1-235:49, 2021.
> > > >
> > > > [5] Eduardo C. Garrido-Merchán and Daniel Hernández-Lobato. 2020. Dealing with categorical and integer-valued variables in Bayesian Optimization with Gaussian processes. Neurocomput. 380, C (Mar 2020), 20–35. https://doi.org/10.1016/j.neucom.2019.11.004
> > > >
> > > > [6] Ricardo Baptista and Matthias Poloczek. Bayesian optimization of combinatorial structures. In International Conference on Machine Learning, pages 462–471. PMLR, 2018.
> > > >
> > > > [7] Changyong Oh, Jakub Tomczak, Efstratios Gavves, and Max Welling. Combinatorial Bayesian optimization using the graph cartesian product. In H. Wallach, H. Larochelle, A. Beygelzimer, F. d'Alché-Buc, E. Fox, and R. Garnett, editors, Advances in Neural Information Processing Systems, volume 32. Curran Associates, Inc., 2019.
> > > >
> > > > [8] Binxin Ru, Ahsan Alvi, Vu Nguyen, Michael A. Osborne, and Stephen Roberts. Bayesian optimisation over multiple continuous and categorical inputs. In Hal Daumé III and Aarti Singh, editors, Proceedings of the 37th International Conference on Machine Learning, volume 119 of Proceedings of Machine Learning Research, pages 8276–8285. PMLR, 13–18 Jul 2020.
> > > >
> > > > [9] Antoine Grosnit, Cedric Malherbe, Rasul Tutunov, Xingchen Wan, Jun Wang, and Haitham Bou Ammar. BOiLS: Bayesian Optimisation for Logic Synthesis. In Proceedings of the 2022 Conference & Exhibition on Design, Automation & Test in Europe, DATE ’22, page 1193–1196, Leuven, BEL, Mar 2022. European Design and Automation Association.
> > > >
> > > > [10] Xingchen Wan, Vu Nguyen, Huong Ha, Binxin Ru, Cong Lu, and Michael A Osborne. Think global and act local: Bayesian optimisation over high-dimensional categorical and mixed search spaces. International Conference on Machine Learning, 2021.
> > > >
> > > > [11] Aryan Deshwal, Sebastian Ament, Maximilian Balandat, Eytan Bakshy, Janardhan Rao Doppa, and David Eriksson. Bayesian optimization over high-dimensional combinatorial spaces via dictionary-based embeddings. CoRR, abs/2303.01774, 2023. doi: 10.48550/arXiv.2303.01774.
> > > >
> > > > [12] Alex Kulesza, Ben Taskar, et al. Determinantal point processes for machine learning. Founda407 tions and Trends® in Machine Learning, 5(2–3):123–286, 2012

---

> > > > > ### Comment · Reviewer_Vgsh · 2023-08-25
> > > > > **Official Comment by Reviewer**
> > > > >
> > > > > I have reviewed the authors' responses to my comments and the corresponding changes made in the revised manuscript.
> > > > >
> > > > > * Discussion of Existing Benchmarking Suites: I appreciate the effort taken to include a detailed comparison, which has strengthened the paper's context within the field.
> > > > >
> > > > > * Inclusion of Other Acquisition Functions: While I recognize the challenges in implementing advanced acquisition functions, I value the authors' transparent and rational explanation.
> > > > >
> > > > > * Detailed Explanation of Benchmark Functions: The updated table in the appendix is informative and provides a good overview.
> > > > >
> > > > > * Choice of Initial Design Size and Number of Function Evaluations: I appreciate the authors clarifying their choices by referencing established practices in prior research.
> > > > >
> > > > > * Statistical Analysis of Experimental Results: Mentioning the conduction of an overall Friedman tests clarifies the evaluation protocol.
> > > > >
> > > > > * Question of "Best Performing Algorithm": Reframing the discussion to focus on the most "robust" algorithm alings more closely with the paper's objectives.
> > > > >
> > > > > * Documentation and Maintenance Plan and Tutorials: The newly added maintenance plan and improved documentation are welcomed additions that will benefit readers and researchers. Thank you very much for providing additional tutorials and notebooks. This makes me confident that the framework will be usable by a wide audience.
> > > > >
> > > > > * Use of Random Search for Acquisition Function Optimization: Implementing a random-search-based acquisition function optimizer serves as an effective baseline which is nice addition.
> > > > >
> > > > > * Support for Other Sampling Strategies: The integration of Sobol sampling is a beneficial addition.
> > > > >
> > > > > In summary, I believe that the authors have addressed the concerns I raised effectively. The revisions not only address the initial concerns but have significantly enhanced the overall quality of the paper.
> > > > >
> > > > > I am therefore increasing my rating.

---

### Official Review · Reviewer_MaRf · 2023-07-21
**Excellent toolkit and benchmark contribution to bayesian optimization**

**Rating:** 8
**Confidence:** 4
**Correctness:** Yes.
**Clarity:** Yes.

**Strengths:**

1. Significance and relevance.

This paper makes significant and fundamental contributions to the field in benchmarking and evaluating BO algorithms. I appreciate the efforts in unifying BO algorithms, building easily accessible tookits for future use, providing various datasets and environments for application, and evaluating the performance of various BO algorithms. I believe the provided tools will greatly help the development of BO algorithms in the field.

2. Quality of research.

This paper is of a high quality and contains fruitful results. The tools are well built; the datasets and environments are comprehensive; the experiments are fruitful. It is also well-written and easy to follow.

**Additional Feedback:**

I don't have big issues in mind. Below are some suggestions that may enhance this paper.

1. Demonstrate use of the framework for new methods.

While the authors did a great job documenting how the framework can be used, it will be nice to also include a few lines of code on how these modules can work with other newly developed ones (or in supplementary material), since this will be helpful for subsequent researchers to use this framework.

2. In figure 4, I was a bit confused at first sight since I thought the goal is to maximize an objective function. It might be helpful to clarify what "best observed value" is.

**Documentation:**

Yes.

**Ethics:**

No.

**Limitations:**

Yes.

**Opportunities For Improvement:**

I don't have particular points for improvement.

**Relation To Prior Work:**

Yes.

**Summary And Contributions:**

This paper presents a new and comprehensive framework for Bayesian optimization (BO), including a library of modules for existing and new BO algorithms, and benchmarks for a wide range of BO algorithms. The toolkits it provides are easy to use and compatible with future new algorithms. It also provides various real-world environments for applying BO to real datasets. The evaluation of algorithms demonstrates the performance of various BO algorithms and points out future directions to pursue.

---

> ### Author Response · Authors · 2023-08-24
>
> We would like to thank the reviewer for their suggestions and feedback. Below are our responses to the comments raised by the reviewer. Note that associated revisions of the PDF are highlighted in blue in the new version.
>
> > Demonstrate use of the framework for new methods.
>
> In the updated version of the MCBO codebase, we provide a set of [guides and tutorials]([official MCBO codebase](https://github.com/huawei-noah/HEBO/tree/master/MCBO/tutorials)) to explain how to integrate new components to the mix-and-match machinery, to run a new method on our proposed set of benchmarks, as well as to visualize the performance of the new method compared to our baselines.
>
> > In figure 4, I was a bit confused at first sight since I thought the goal is to maximize an objective function. It might be helpful to clarify what "best observed value" is.
>
> We thank the reviewer for this comment. We have now updated the captions of Figures 4 and 8 to clarify that the aim is to minimize the black-box function in all considered tasks.
> Additionally, we clearly state in the library [README.md](https://github.com/huawei-noah/HEBO/tree/master/MCBO#simple-optimization-example) that the optimizers of the library are implemented to minimize the black-box function.

---

### Official Review · Reviewer_m29E · 2023-07-22
**Review of: Framework and Benchmarks for Combinatorial and Mixed-variable Bayesian Optimization**

**Rating:** 7
**Confidence:** 3

**Strengths:**

The decomposition of existing MCBO methods into their three constituent parts and the implementation of multiple variants of each part is a valuable contribution in terms of the structure of an MCBO software library. The implementation of multiple tasks for use in benchmarking is also a useful contribution. In terms of the benchmark results, the authors report some novel findings regarding the performance of new mix-and-match model variants as well as non-BO methods.

**Additional Feedback:**

* The authors should consider giving a more explicit definition of a combinatorial space, for example, by defining the set from which x is drawn in equation (1).

* Line 72: "Secondly, papers introducing a solution for a single MCBO primitive often forget to benchmark against baselines that use the same methods for the remaining primitives, failing to fully highlight the merits of their proposed solution in a controlled setting. " -- The authors should supply references to papers exhibiting this issue to support this claim.

* Line 138: "their traceability" -- Should this be tracktability?

* Line 202: suit -> suite

* Line 211: "implementing methods evaluate that calls the black-box get_search_space that returns an instance of SearchSpace specifying the black-box domain." -- Consider revising to improve clarity.

* Line 224: "On acquisition function" -> One acquisition function

* Line 322: "Finally, Fig. 2 (right) pleads for the use of a TR in combinatorial BO." -- This is a fairly colloquial turn of phrase. Consider revising.

* Line 360: "We get the quality of a GP fit on a given task and seed by conditioning the GP on the first 150 points coming from our GA run and computing the log-likelihood of the last 50 black-box values" -- Please confirm that the same set of points is being conditioned on and evaluated for all methods being compared.

* Line 363: "the GP log-lihood" ->  "the GP log-likelihood"

**Clarity:**

The paper is generally well written. Some typos and minor corrections are included in the detailed comments. One potential issue with the paper is that the authors make extensive use of short and quite similar acronyms for naming methods and method components. Consider for example, this excerpt below from lines 224-228 that includes in two sentences the acronyms EI, PI, LCB, TS, LHS, LS, GA, SA, HC, and MAB. This is typical of the writing style in the paper. While it is an efficient style in terms of space, the authors should realize that it will negatively impact the readability of the paper for readers who are not already familiar with all of the acronyms. A potential solution is to include a list of all acronyms in alphabetical order at the start of the appendix and to let readers know that it is available there.

"2) On acquisition function side, we include the widely used EI, PI, LCB for the GPs, and TS for LHS. 3) Furthermore, our library includes various acquisition optimizers, such as LS [1], GA [6], SA [5], and interleaved search (IS) alternating between HC or MAB for combinatorial variables and gradient-descent steps for numeric variables as developed for oCaBO [7] and Casmopolitan [2]."

**Correctness:**

The evaluation methods used in the benchmark experiments appear to be correct. Some related questions and comments are included here:

* Line 308: "On figures displaying evolution of ranks, we show the mean rank in solid line, the standard error with respect to tasks and seeds as a shaded area." -- Please clarify whether the distribution of ranks about the mean rank is sufficiently symmetric that using a standard error region is an adequate representation of variability about the mean rank.

* Line 364: "As expected, Fig. 5 shows a positive correlation ..." -- It is fairly difficult to assess the correlation level from the figures as the variables are discrete. It may be preferable to report the actual correlation values in a figure.

**Documentation:**

The core repository includes a function for reproducing all of the results in the paper. Due to the two week run time, the reviewer did not attempt to verify this functionality.

As noted above, a significant limitation of the front page documentation is the lack of inclusion of documentation and examples showing how to add a new method to the library, evaluate it against the full benchmark set of tasks, and produce corresponding figures.

**Ethics:**

I do not have ethical concerns about this submission.

**Limitations:**

The appendix includes a discussion of the hardware used and the compute time required to obtain the presented results. This information should be moved to the main paper as it sheds light on what would be required to reproduce the results reported in the paper (likely weeks of computation). It would also be helpful to break this down in terms of the run time required for the experiment needed to benchmark a single new method as this is the most likely use case for a user of this framework.

**Opportunities For Improvement:**

The authors should be more clear about whether they are advocating that the the currently implemented set of tasks is a robust benchmark. The main utility of publishing benchmark results is that future work can run new algorithms against the same set of tasks to determine whether new methods have improved overall performance. To that end, the provided documentation and examples included in the code repository should be updated to facilitate the addition of new methods and new method components. The repository should also be updated to include CSV or JSON files containing the numerical results of all of the evaluations performed by the authors along with code for reproducing the figures in the paper with documentation of how to add results for additional methods.

There are more minor presentation issues described in the detailed comments.

**Relation To Prior Work:**

The authors mention a significant amount of related work without going in to any detail as they are simply enumerating previously published methods and method components for the most part, which is fine. The one point that could use additional attention is describing what fraction of results in prior work are covered by the tasks included in the proposed library.

**Summary And Contributions:**

This paper presents a library for supporting the benchmarking of methods for combinatorial and mixed-variable Bayesian optimization (MCBO). It includes functionality for defining tasks and method components including the Surrogate Model, Acquisition Function and Acquisition Optimization Method. The decomposition of existing MCBO methods into these three components allows the authors to implement a modest sized collection of versions of each component following a common API, and then mix and match them into a large number of MCBO method variants that includes several published algorithms as well as many previously unpublished combinations. The authors also include several non-BO based baseline methods.

The authors perform evaluations on multiple tasks using many combinations of the the model components. They show that some of the novel combinations perform at least as well as some of the previously published methods. While the proposed software has some limitations from the perspective of evaluating new methods, this is mostly a documentation issue that can be fixed by the authors. More clarity is also needed regarding whether the authors consider the currently implemented tasks to be a sufficiently exhaustive benchmark such that researchers can reliably use it to determine the merits of newly proposed methods.

---

> ### Author Response · Authors · 2023-08-24
>
> We thank the reviewer for their thoughtful and constructive comments! Below are our responses to the individual comments made by the reviewer. All revisions to the PDF we brought following the reviewers' comments are in blue.
>
> > The authors should be more clear about whether they are advocating that the currently implemented set of tasks is a robust benchmark.
>
> We advocate that the currently implemented set of tasks is a robust benchmark due to its diversity and focus on practical real-world problems. The implemented tasks encompass various domains, including hyperparameter tuning, antibody design, Electronic Design Automation, RNA inverse folding, and synthetic tasks. All these tasks cover a broad spectrum of problem types and complexities and have their own challenges. By conducting experiments across such a diverse set of tasks, we aimed to strengthen the validity and generalizability of our findings. We believe that this multifaceted approach leads to more robust and generalisable conclusions than scenarios where only a single family of tasks is used for benchmarking - which can still be useful for important black-box families.
>
> Furthermore, while we acknowledge that in our experiments we have not focused on very-high dimensional problems with hundreds or thousands of variables, the 21 synthetic SFU test functions we have implemented can all be instantiated to an arbitrary number of dimensions, and would therefore allow the study of such problems using our framework.
>
> > To that end, the provided documentation and examples included in the code repository should be updated to facilitate the addition of new methods and new method components.
>
> We have put a significant effort into providing a structured document along with notebook tutorials in order to ease the integration of new modules (surrogate models, acquisition optimizers, etc.) in our MCBO library, which we explain how to do step by step.
> This new documentation is available in the [official MCBO codebase](https://github.com/huawei-noah/HEBO/tree/master/MCBO/tutorials)
>
> > The repository should also be updated to include CSV or JSON files containing the numerical results of all of the evaluations performed by the authors along with code for reproducing the figures in the paper with documentation of how to add results for additional methods.
>
> We provide a [zip file data.zip](https://github.com/huawei-noah/HEBO/blob/master/MCBO/paper_results/data.zip) that compresses the two CSV files (one for combinatorial, one for mixed runs) containing the numerical results of all our optimization experiments. We also provide [two notebooks](https://github.com/huawei-noah/HEBO/blob/master/MCBO/paper_results/) that we created to make the figures presented in our paper.
>
> Additionally, a [section](https://github.com/huawei-noah/HEBO/tree/master/MCBO/tutorials#evaluate-a-new-bo-algorithm-on-the-mcbo-tasks) of our documentation explains how to run a new algorithm on our set of benchmarks, and we provide a [notebook](https://github.com/huawei-noah/HEBO/blob/hebo/MCBO/tutorials/tuto_results_viz.ipynb) to show how to visualize the results of the new runs, as well as how to integrate the results of the baselines we already run to the new graph of results.
>
> > The appendix includes a discussion of the hardware used and the compute time required to obtain the presented results...
>
> Taking advantage of the extra page, we have now moved this paragraph to the main paper.

---

> > ### Author Response · Authors · 2023-08-24
> >
> > > It would also be helpful to break this down in terms of the run time required for the experiment needed to benchmark a single new method as this is the most likely use case for a user of this framework.
> >
> > We thank the reviewer for this suggestion.
> > To provide an estimation of the runtime of evaluating a new method, we have added [in the documentation a table](https://github.com/huawei-noah/HEBO/tree/master/MCBO/tutorials#evaluate-a-new-bo-algorithm-on-the-mcbo-tasks) specifying for each task how long it takes on average to query the black-box 200 times sequentially (so not taking into account the time to suggest a new query point).
> > Then, the runtime of benchmarking a single new method on our benchmarks is also a function of the BO primitives the new method employs. For this reason, we now include a function that takes as input an instance of a task, an optimizer and the total black-box evaluation budget and returns an approximate total runtime estimate to get an optimization trajectory for a single seed.
> > This is done by just querying the black-box once to approximate the time taken for each black-box evaluation (or by providing this value to the function if it is known, as for our benchmark tasks), and by querying the `suggest` method of the optimizer for different dataset sizes.
> > Then, a linear interpolation gives a rough estimate of the total runtime of the BO loop for all iterations. Examples of estimated and actual runtimes for the MCBO baselines can be seen in the table presented in this [section of the guide](https://github.com/huawei-noah/HEBO/tree/hebo/MCBO/tutorials#runtime-estimation).
> > Knowing the runtime of each black-box of the benchmark, and being able to estimate quickly the suggestion runtime of a new method, one can quickly estimate the total runtime by taking into account the degree of parallelism available.
> >
> > > Please clarify whether the distribution of ranks about the mean rank is sufficiently symmetric that using a standard error region is an adequate representation of variability about the mean rank.
> >
> > In order to clarify this, we have generated the violin plots (accessible via [this link](https://github.com/huawei-noah/HEBO/tree/hebo/MCBO/paper_results/images/rank_symmetry#rank-distribution-symmetry)) showing the empirical distribution or the ranks achieved after 200 optimization steps (so at the end of each run) by the optimizers of Figure 3 and Figure 7, and by the groups of optimizers for Figure 2 and Figure 6.
> > They overall appear to be approximately symmetric, notably for the
> > ones aggregating results per model, acquisition optimizer, or use of a trust region.
> > As for the ranks of individual optimizers, it is true that some distributions appear more skewed (e.g. for COMBO on combinatorial tasks, or Genetic Algorithm for mixed tasks), but showing the standard error of the mean rank still provides information on the variance of the optimizer's performance. Note that in addition to the mean rank, we provide the regret plots per task, which sheds light on the magnitude and skewness of the variation of the average performance of each optimizer.
> >
> > >  It is fairly difficult to assess the correlation level from the figures as the variables are discrete. It may be preferable to report the actual correlation values in a figure.
> >
> > We thank the reviewer for pointing this out. Our updated figure now explicitly indicates the actual correlation value for each plot.
> >
> > >  A potential solution is to include a list of all acronyms in alphabetical order at the start of the appendix and to let readers know that it is available there.
> >
> > We thank the reviewer for the suggestion and now include a full list of the abbreviations used in the paper at the start of the appendix and mention this at the end of the introduction.

---

> > > ### Author Response · Authors · 2023-08-24
> > >
> > > > ... The one point that could use additional attention is describing what fraction of results in prior work are covered by the tasks included in the proposed library.
> > >
> > > We appreciate the reviewer's keen observation regarding the related work section, and we thank them for their thoughtful feedback. In response, we wish to shed light on the coverage of tasks within our proposed framework as compared to prior work on mixed-variable and combinatorial Bayesian Optimization [1, 2, 3, 4, 5, 6, 7].
> > >
> > > Our framework encompasses 43% of all previously explored tasks with open-source implementations and 60% of tasks involving non-binary variables. While we have initially omitted tasks with binary-only variables, including binary quadratic programming [1], contamination control [1, 5], Ising sparsification [1, 2], low auto-correlation binary sequences [6] and weighted maximum satisfiability [2, 5, 6], we plan to integrate them into our framework in the near future. However, before then, we are prioritising the integration of the remaining tasks defined over nominal variables, which are: adversarial attack on a convolutional neural network [5], neural architecture search [2, 3] and Func-XC [3, 5], where X is the number of combinatorial variables. We also would like to mention that even though our framework does not currently encompass all tasks explored by prior work, our benchmarking suite includes 20 additional synthetic test functions adaptable to arbitrary combinations of variable types and dimensionalities and extends current application venues of Bayesian Optimisation to MIG sequence optimisation, AIG sequence and parameter optimisation, and RNA inverse folding, all of which are important real-world tasks.
> > >
> > > Furthermore, we acknowledge that the estimate of the fraction of task-method results we cover from prior work is likely to be lower than our task coverage. The reason for this is that [3, 5, 2, 6] all benchmark against either SMAC [8], TPE [9], or both, and our library does not currently implement these baselines. The reason for this is that we prioritized the inclusion of contemporary algorithms that have been shown to consistently outperform these older methods.
> > >
> > > > The authors should consider giving a more explicit definition of a combinatorial space, for example, by defining the set from which x is drawn in equation (1).
> > >
> > > Following this advice, we now include a definition of the set from which x is drawn following equation 1.
> > >
> > > > "Secondly, papers introducing a solution for a single MCBO primitive often forget to benchmark ... failing to fully highlight the merits of their proposed solution in a controlled setting. " -- The authors should supply references to papers exhibiting this issue to support this claim.
> > >
> > > We have now added citations supporting this claim to both the abstract and line 72 of our manuscript.
> > >
> > > >  "Finally, Fig. 2 (right) pleads for the use of a TR in combinatorial BO." -- This is a fairly colloquial turn of phrase. Consider revising.
> > >
> > > We have now modified this in the revised version of our manuscript.
> > >
> > > > "We get the quality of a GP fit on a given task and seed by conditioning the GP ... and computing the log-likelihood of the last 50 black-box values" -- Please confirm that the same set of points is being conditioned on and evaluated for all methods being compared.
> > >
> > > Indeed, for a given task and seed,  the points coming from the GA are the same across all the methods being compared to make the comparison fair. We have clarified this point in the revised version of the paper.

---

> > > > ### Author Response · Authors · 2023-08-24
> > > >
> > > > References:
> > > >
> > > > [1] Ricardo Baptista and Matthias Poloczek. Bayesian optimization of combinatorial structures. In International Conference on Machine Learning, pages 462–471. PMLR, 2018.
> > > >
> > > > [2] Changyong Oh, Jakub Tomczak, Efstratios Gavves, and Max Welling. Combinatorial Bayesian optimization using the graph cartesian product. In H. Wallach, H. Larochelle, A. Beygelzimer, F. d'Alché-Buc, E. Fox, and R. Garnett, editors, Advances in Neural Information Processing Systems, volume 32. Curran Associates, Inc., 2019.
> > > >
> > > > [3] Binxin Ru, Ahsan Alvi, Vu Nguyen, Michael A. Osborne, and Stephen Roberts. Bayesian optimisation over multiple continuous and categorical inputs. In Hal Daumé III and Aarti Singh, editors, Proceedings of the 37th International Conference on Machine Learning, volume 119 of Proceedings of Machine Learning Research, pages 8276–8285. PMLR, 13–18 Jul 2020.
> > > >
> > > > [4] Antoine Grosnit, Cedric Malherbe, Rasul Tutunov, Xingchen Wan, Jun Wang, and Haitham Bou Ammar. BOiLS: Bayesian Optimisation for Logic Synthesis. In Proceedings of the 2022 Conference & Exhibition on Design, Automation & Test in Europe, DATE ’22, page 1193–1196, Leuven, BEL, Mar 2022. European Design and Automation Association.
> > > >
> > > > [5] Xingchen Wan, Vu Nguyen, Huong Ha, Binxin Ru, Cong Lu, and Michael A Osborne. Think global and act local: Bayesian optimisation over high-dimensional categorical and mixed search spaces. International Conference on Machine Learning, 2021.
> > > >
> > > > [6] Aryan Deshwal, Sebastian Ament, Maximilian Balandat, Eytan Bakshy, Janardhan Rao Doppa, and David Eriksson. Bayesian optimization over high-dimensional combinatorial spaces via dictionary-based embeddings. CoRR, abs/2303.01774, 2023. doi: 10.48550/arXiv.2303.01774.
> > > >
> > > > [7] Henry B. Moss, David S. Leslie, Daniel Beck, Javier Gonzalez, and Paul Rayson. BOSS: Bayesian optimization over string spaces. In NeurIPS, 2020.
> > > >
> > > > [8] Hutter, F., Hoos, H.H., Leyton-Brown, K. (2011). Sequential Model-Based Optimization for General Algorithm Configuration. In: Coello, C.A.C. (eds) Learning and Intelligent Optimization. LION 2011. Lecture Notes in Computer Science, vol 6683. Springer, Berlin, Heidelberg. https://doi.org/10.1007/978-3-642-25566-3_40
> > > >
> > > > [9] Bergstra, J., Bardenet, R., Bengio, Y., & Kegl, B. (2011). Algorithms for Hyper-Parameter Optimization. In Proceedings of the 24th International Conference on Neural Information Processing Systems (pp. 2546–2554). Curran Associates Inc..

---

> > > > > ### Comment · Reviewer_m29E · 2023-08-30
> > > > > **Author response**
> > > > >
> > > > > I thank the authors for their detailed response. My primary concerns about documentation and extensibility have been addressed as have the other issues I raised. I continue to recommend that the paper be accepted.

---

### Author Response · Authors · 2023-08-24
**General comment**

We thank all reviewers for their time and comments. Based on all the feedback, we are pleased to inform that we have put significant efforts into improving our paper and framework. Namely, we now:
- Have updated our related work section to distinguish our work from existing mixed-variable and combinatorial optimization benchmarking suites.
- Provide a lexicon defining all the abbreviations we use in our paper at the beginning of the appendix.
- Provide details on the domain of each benchmark in Table 2 of the appendix.
- Have improved our documentation.
- Provide [tutorials](https://github.com/huawei-noah/HEBO/tree/master/MCBO/tutorials) on how to:
  - Include and optimize a custom black-box function.
  - Add a new surrogate model to our framework.
  - Add a new acquisition function to our framework.
  - Add a new acquisition function optimizer to our framework.
  - Evaluate a new MCBO algorithm on our benchmarks.
  - Estimate the runtime of an algorithm on a task.
- Provide a [Notebook](https://github.com/huawei-noah/HEBO/blob/master/MCBO/tutorials/tuto_results_viz.ipynb) demonstrating how to visualise results for a new algorithm and how to compare them to the baselines we have run.
- Provide an implementation of a [random search acquisition optimizer](https://github.com/huawei-noah/HEBO/blob/master/MCBO/mcbo/acq_optimizers/random_search_acq_optimizer.py) that is integrated with the `BoBuilder` class.
- Provide an implementation of Sobol sampling to generate the initial points suggested by a BO instance, and also integrated with the `BoBuilder` class.

If any of the reviewers have any further questions, we would be pleased to answer them.

---

### Decision · Program_Chairs · 2023-09-22

**Decision:**

Accept (Poster)

**Comment:**

Good paper to accept.